# Finally Rank-Breaking Conquers MNL Bandits: Optimal and Efficient Algorithms for MNL Assortment

**Aadirupa Saha** [*]
Department of Computer Science
University of Illinois, Chicago
aadirupa.saha@gmail.com

**Pierre Gaillard**
Univ. Grenoble Alpes, Inria, CNRS
Grenoble INP, LJK, 38000 Grenoble, France
pierre.gaillard@inria.fr

## Abstract

We address the problem of active online assortment optimization problem with preference feedback, which is a framework for modeling user choices and subsetwise utility maximization. The framework is useful in various real-world applications including ad placement, online retail, recommender systems, and fine-tuning language models, amongst many others. The problem, although has been studied in the past, lacks an intuitive and practical solution approach with simultaneously efficient algorithm and optimal regret guarantee. E.g., popularly used assortment selection algorithms often require the presence of a 'strong reference' which is always included in the choice sets, further they are also designed to offer the same assortments repeatedly until the reference item gets selected—all such requirements are quite unrealistic for practical applications. In this paper, we designed efficient algorithms for the problem of regret minimization in assortment selection with *Multinomial Logit* (MNL) based user choices. We designed a novel concentration guarantee for estimating the score parameters of the PL model using '*Pairwise Rank-Breaking*', which builds the foundation of our proposed algorithms. Moreover, our methods are practical, provably optimal, and devoid of the aforementioned limitations. Empirical evaluations corroborate our findings and outperform the existing baselines.

## 1 Introduction

Studies have shown that it is often easier, faster and less expensive to collect feedback on a relative scale rather than asking ratings on an absolute scale. E.g., to understand the liking for a given pair of items, say (A,B), it is easier for the users to answer preference-based queries like: "Do you prefer Item A over B?", rather than their absolute counterparts: "How much do you score items A and B in a scale of [0-10]?" (Musallam et al., 2004; Kahneman & Tversky, 1982). Due to the widespread applicability and ease of data collection with relative feedback, learning from preferences has gained much popularity in the machine-learning community, especially the active learning literature which has applications in Medical surveys, AI tutoring systems, Multi-player sports/games, or any real-world systems that have ways to collect feedback in terms of preferences. The problem is famously studied as the *Dueling-Bandit* (DB) problem in the active learning community Yue et al. (2012); Ailon et al. (2014); Zoghi et al. (2014a;b; 2015), which is an online learning framework for identifying a set of 'good' items from a fixed decision-space (set of items) by querying preference feedback of actively chosen item-pairs. Consequently, the generalization of Dueling-Bandits, with *subset-wise* preferences has also been developed into an active field of research. For instance, applications like Web search, language models, online shopping, recommender systems (e.g. Youtube, Netflix, Google News/Maps, Spotify) typically involve users expressing preferences

---

[*]Part of the work was done while author was at Apple, ML Research

by choosing one result (or a handful of results) from a subset of offered items and often the objective of the system is to identify the 'most-profitable' subset to offer to their users. The problem, popularly termed as 'Assortment Optimization' is studied in many interdisciplinary literature, e.g. Online learning and bandits Bengs et al. (2021a), Operations research Talluri & Van Ryzin (2004); Agrawal et al. (2019), Game theory Chatterji et al. (2021), RLHF Christiano et al. (2017); Ouyang et al. (2022), to name a few.

**Problem (Informal): Active Optimal Assortment (AOA)** Active Assortment Optimization (a.k.a. Utility Maximization with Subset Choices) Berbeglia & Joret (2016); Agrawal et al. (2019); Désir et al. (2016b;a) is an active learning framework for finding the 'optimal' profit-maximizing subset. Formally, assume we have a decision set of $[K] := \{1, 2, \ldots K\}$ of $K$ items, with each item being associated with the score (or utility) parameters $\boldsymbol{\theta} := (\theta_1, \theta_2, \ldots, \theta_K)$ (without loss of generality assume $\theta_1 \geq \theta_2 \geq \ldots \geq \theta_K \geq 0$). At each round $t = 1, 2, \ldots$, the learner or the algorithm gets to query an assortment (typically subsets containing up to $m$-items) $S_t \subseteq [K]$, upon which it gets to see some (noisy) relative preferences across the items in $S_t$, typically generated according to an underlying Multinomial Logit (MNL) model with parameters $\boldsymbol{\theta}$ (1). Further, to allow the event where no items are selected, we also model a No-Choice (NC) item, indexed by item-0, with PL parameter $\theta_0 \in \mathbb{R}_+$.

**(Objective 1.) Top-$m$:** identify the top-$m$ item-set: $\{\theta_1, \ldots, \theta_m\}$, for some $m \in [1, K]$.

**(Objective 2.) Wtd-Top-$m$:** A more general objective could also consider a weight (or price) $r_i \in \mathbb{R}_+$ associated with the item $i \in [K]$, and the goal could be to identify the assortment (subset) with maximum weighted utility [1], as detailed in Sec. 2.

**Related Works and Limitations:** As stated above, the problem of AOA is fundamental in many practical scenarios, and thus widely studied in multiple research areas, including Online ML/learning theory and operations research.

• In the Online ML literature, the problem is well-studied as *Multi-Dueling Bandits* Sui et al. (2017); Brost et al. (2016), or Battling Bandits Saha & Gopalan (2019a; 2018); Bengs et al. (2021b), which is an extension of the famous *Dueling Bandit* problem Zoghi et al. (2014b;a). The main limitation of this line of work is the lack of practical objectives, which either aim to identify the 'best-item' $1(= \arg\max_{i \in [K]} \theta_i)$ within a PAC (probably approximately correct) framework Saha & Gopalan (2019b); Chen et al. (2017; 2018); Ren et al. (2018) or quantifying regret against the best items Saha & Gopalan (2019a); Bengs et al. (2022). Note the latter actually leads to the optimal subset choice of repeatedly selecting the optimal item, $\arg\max_i \theta_i$, $m$ times, i.e. $(1, 1, \ldots 1)$, which is unrealistic from the viewpoint of real-world system design. Selecting an assortment of distinct top-$m$ items (Top-$m$-AOA) or maximum expected utility (Wtd-Top-$m$-AOA) makes more sense.

• On the other hand, a similar line of the problem has been studied in operations research and dynamic assortment selection literature, where the goal is to offer a subset of items to the customers in order to maximize expected revenue. The problem has been studied under different user choice models, e.g. PL or Multinomial-Logit models (Agrawal et al., 2019), Mallows and mixture of Mallows (Désir et al., 2016a), Markov chain-based choice models (Désir et al., 2016b), single transition model (Nip et al., 2017) etc. While these works indeed consider a more practical objective of finding the best assortment (subset) with the highest expected utility for a regret minimization objective, (1) a major drawback in their approach lies in the algorithm design which *requires to keep on querying the same set multiple times*, e.g. Agrawal et al. (2019); Ou et al. (2018); Chen et al. (2021); Agrawal et al. (2017). Such design techniques could be impractical to be deployed in real systems where users could easily get annoyed if the same items are shown again and again. For example, in ad-placement, music/movies/news/tweets/reels recommendations, offering the same assortment could increase user dissatisfaction and disengagement.

(2) The second major drawback of this line of work lies in the *structural assumption of their underlying choice models which requires the existence of a reference/default item, that*

---

[1]This is equivalent to finding the set with maximum expected revenue when $r_i$s represents the price of item $i$ Agrawal et al. (2019)

*needs to be part of every assortment $S_t$*. This leads to assuming a No-Choice item, typically denoted as item-0, which is a default choice of any assortment $S_t$. Further a stronger and more unrealistic assumption lies in the fact that they require to assume that the above pivot is stronger than the rest of the $K$ items, i.e. $\theta_0 \geq \max_{i \in [K]} \theta_i$, i.e. the No-Choice (NC) action is the most likely outcome of any assortment $S_t$. This is often unrealistic, e.g., during user interactions with language models, or online shopping, or maps recommendation, users typically make choices as the user needs to commute or book a flight and a NC action is highly improbable, e.g. in recommender systems like YouTube, Spotify, Netflix or even Yahoo News, users typically make choices as they actually wanted to consume a video, new article or song, etc. Similarly, in shopping recommendations like flights (Expedia or Google flights), hotel (Booking.com), restaurants (Grubhub), or Google Maps recommendations, NC is unlikely. In fact, in some applications, NC might not even be an available (feasible) option, e.g., while interacting with ChatGPT/ Gemini, the language model often requests the users to definitively select one outcome in order to proceed with the thread; Similarly, in robotics applications, training of autonomous vehicles, or more generally in any preference-based RL (a.k.a. PbRL) applications, the teacher/ demonstrator/ human feedback provider must choose an option out of the multiple options (or RL trajectories) towards training the RL policy. Consequently, such assumption limits the use in real-systems. In the existing literature Agrawal et al. (2019); Oh & Iyengar (2019); Agrawal et al. (2017); Grant & Leslie (2023), such assumptions are primarily adapted solely for theoretical needs, precisely for maintaining concentration bounds of the PL parameters $\boldsymbol{\theta}$, and hence not well justified from a practical viewpoint.

Agrawal et al. (2019) is the classical MNL-Assortment work with MNL-UCB: The idea is to estimate the true PL parameter $\boldsymbol{\theta} = (\theta_1, \ldots, \theta_K)$s by repeatedly querying the same set (i.e. assortment $S_t$) multiple times and keeping a count of the average number of times an item $i \in [K]$ is selected until no items (NC) are selected. They further maintain a UCB of the estimated PL parameters, $(\widehat{\theta}_1, \ldots, \widehat{\theta}_K)$, and the assortment of the next phase optimistically based on the UCB estimates. The process is repeated until time step $T$. Agrawal et al. (2017) is a follow-up work of Agrawal et al. (2019) from the same group of authors and hence their algorithm MNL-TS is almost the same as MNL-UCB above, with the exception of using Thompson Sampling (TS) with Beta posteriors, instead of the UCB estimates. The regret guarantee and set of restrictive assumptions imposed on the MNL model is also identical to that of Agrawal et al. (2019). The objective of Grant & Leslie (2023) is slightly different from that of above two as their objective is 'learning to rank' (LTR), i.e. to find the best ordered subset based on some position bias $\lambda_i > 0$ for position $i \in [m]$. More generally, their preference model is different which assumes the probability of the item played at position-$k$ getting selected (or clicked) for any $m$-length ordered set $S = (S(1), \ldots, S(m))$ is given by $P(k \mid S) := \lambda_k \theta_{S(k)} / (\theta_0 + \sum_{j \in [m]} \lambda_j \theta_{S(j)})$, where as usual $\theta_0$ is the score parameter of the no-choice item. We summarize these existing works in Table 1.

Some recent developments also generalized the AOA problem to linear MNL scores to incorporate large actions embedded in $d$-dimension Zhang & Ji (2019); Zhang & Sugiyama (2024); Oh & Iyengar (2019), however, their approaches are either limited to the above restrictions or suffer sub-optimal regret guarantees without those assumptions (e.g. the regret bound of Oh & Iyengar (2019) is $O(d^{3/2}\sqrt{T})$ which is suboptimal by a $d$-factor). Considering the above limitations of the AOA literature, we set to answer two questions:

(1) Can we consider a general AOA model where the default item, like the NC item defined above, is not necessarily the strongest one, i.e. $\theta_0 \geq \max_{i \in [K]} \theta_i$?
(2) Can we design a practical and regret optimal algorithm for the AOA framework, without needing to play the same repetitive actions and yet converge to the optimal assortment?

| Work | Framework | Assume $\theta_0 = \theta_{\max} = 1$ | Regret |
|---|---|---|---|
| Our (Alg. 1) | MNL model (Obj. 2) | No | $\sqrt{\min\{\theta_{\max}, K\}KT\log T}$ |
| Agrawal et al. (2019) (Thm 1) | MNL model (Obj. 2) | Yes | $\sqrt{KT\log T}$ |
| Agrawal et al. (2019) (Thm 4) | MNL model (Obj. 2) | No | $\sqrt{\theta_{\max}KT\log T}$ |
| Agrawal et al. (2017) | MNL model (Obj. 2) | Yes | $\sqrt{KT\log(mT)}$ |
| Grant & Leslie (2023) | MNL model with constraints (Obj. 2) | No | $\sqrt{\frac{KT}{\min_i r_i}}\log T$ |

Table 1: Our Contribution vs the Existing Results in the $K$-armed MNL-Assortment literature. The regret statements include only the leading asymptotic term, ignoring constant factors.

**Contributions**  We answer these questions in the affirmative and present best of all scenarios. We design practical algorithms on practical AOA framework with practical objectives–Unlike the existing approaches of the AOA, literature Agrawal et al. (2019); Chen et al. (2021), we do not have to keep playing the same assortment multiple times, neither require a strongest default item (like NC satisfying $\theta_0 \geq \max_{i \in [K]} \theta_i$). Moreover, our objectives do not require us to converge to a multiset of replicated arms like $(1, 1, \ldots 1)$, but converge to the utility-maximizing set of distinct items. We list our contributions below:

1. **A General AOA Setup:** We work with a general problem of AOA for PL model, which requires no additional structural assumption of the $\boldsymbol{\theta}$ parameters such as $\theta_0 \geq \max_i \theta_i$, unlike the existing works. We designed algorithms for two separate objectives Top-$m$ and Wtd-Top-$m$ as discussed above (Sec. 2).

2. **Efficient and Optimal Algorithm using *Rank-Breaking* MNL-Parameter Estimation:**  In Sec. 3, we give a practical, efficient and optimal algorithm for MNL Assortment (up to log factors and the magnitude of $\theta_{\max}$). The regret bound of our algorithm AOA-RB$_{\text{PL}}$ (Alg. 1) yields $\tilde{O}(\sqrt{KT})$ regret for both Top-$m$ and Wtd-Top-$m$ objective. Our algorithms use a novel parameter estimation technique for discrete choice models based on the concept of *Rank-Breaking* (RB) which is one of our key contributions towards designing the efficient and optimal algorithm. This enables our algorithm to perform optimally without requiring the No-Choice item to be the strongest. Appendix A details the key concept of our parameter estimation technique exploiting the concept of RB. Our resulting algorithm plays optimistically based on the UCB estimates of PL parameters and does not require repeating the same subset multiple times, justifying our title.

3. **Improvement with Adaptive Pivots:**  In Sec. 4, we refine the performance of our algorithm by employing the novel idea of 'adaptive pivots' (a reference item) and proposed AOA-RB$_{\text{PL}}$-Adaptive. Performance-wise this removes the asymptotic dependence on $\theta_{\max} = \max_i \theta_i/\theta_0$ in the regret analysis. This enables the algorithm to work effectively in scenarios where the No-Choice item is less likely to be selected, i.e., $\theta_{\max} \gg 1$. This leads to a huge improvement in our experiments, especially in the range of low $\theta_0$, where AOA-RB$_{\text{PL}}$-Adaptive drastically outperforms over the existing baseline. Comparison of our regret bound with existing work is detailed in Table 1.

4. **Emperical Analysis.**  Finally, we corroborate our theoretical results with empirical evaluations (Sec. 5), which certify our superior performance in the general AOA setups.

It is also worth mentioning that our proposed algorithm and their respective regret analysis could be extended to any general random utility (RUM) based preference models Soufiani et al. (2014); Saha & Gopalan (2020), as explained in Rem. 1, the techniques. However, to keep the focus on the AOA problem and ease the presentation, we stick to the special case of MNL choice model based preferences.

## 2  Problem Setup

We write $[n] = \{1, 2, ..., n\}$ and $\mathbb{1}\{\cdot\}$ denotes the indicator function. The symbol $\lesssim$, employed in the proof sketches, represents a coarse inequality.

We consider the sequential decision-making problem of Active Optimal Assortment (AOA), with preference/choice feedback. Formally, the learner is given $[K]$, a finite set of $K$ items

($K > 2$). At each decision round $t = 1, 2, \ldots$, the learner selects a subset $S_t \subseteq [K]$ of up to $m$ items, and receives some (stochastic) feedback about the item preferences of $S_t$, drawn according to some unknown underlying MNL choice model (1) with parameters $\boldsymbol{\theta} = (\theta_1, \theta_2, \ldots, \theta_K) \in \mathbb{R}_+^K$. We assume $\theta_1 \geq \theta_2 \geq \ldots \geq \theta_K$ without loss of generality. An interested reader may check App. A.1 for a detailed discussion on PL models. Given any assortment $S_t$ we also consider the possibility of 'no-selection' of any items given an $S_t$. Following the literature of Agrawal et al. (2019), we model this mathematically as a No-Choice (NC) item, indexed by item-0, and its corresponding PL utility parameter $\theta_0$. Unlike most existing literature on assortment selection, we are not assuming $\theta_0 \not\geq \max_{i \in [K]} \theta_i$. Further, since the PL model is scale independent, we set $\theta_0 = 1$ and scale the rest of the PL parameters.

**Feedback model** The feedback model formulates the information received (from the 'environment') once the learner plays a subset $S_t \subseteq [K]$ of at most $m$ items. Given $S_t$ we consider the algorithm receives a winner feedback (or index of an item) $i_t \in S_t \cup \{0\}$, drawn according to the underlying PL choice model as:

$$\mathbb{P}(i_t = i | S_t) = \theta_i / \big(\theta_0 + \textstyle\sum_{j \in S_t} \theta_j\big), \quad \forall i \in S_t. \tag{1}$$

We consider the following two objectives for the learner:

**1. Top-$m$-Ojective.** One simple objective could be to identify the top-$m$ item-set: $\{\theta_1, \ldots, \theta_m\}$, for some $m \in [1, K]$. The performance of the learner can be captured by minimizing the following regret:

$$Reg_T^{\texttt{top}} := \sum_{t=1}^{T} \frac{\Theta_{S^*} - \Theta_{S_t}}{m}, \quad \text{where} \quad S^* := \operatorname*{argmax}_{S \subseteq [K]: |S| = m} \left\{ \Theta_S := \sum_{i \in S} \theta_i \right\}.$$

**2. Wtd-Top-$m$-Objective.** Here, each item-$i$ is associated with a weight (for example price) $r_i \in \mathbb{R}_+$, and the goal is to identify the set of size at most $m$ with maximum weighted utility. One could measure the regret of the learner as:

$$Reg_T^{\texttt{wtd}} := \sum_{t=1}^{T} (\mathcal{R}(S^*, \boldsymbol{\theta}) - \mathcal{R}(S_t, \boldsymbol{\theta})), \text{ where } \mathcal{R}(S, \boldsymbol{\theta}) := \sum_{i \in S} \frac{r_i \theta_i}{\theta_0 + \sum_{j \in S} \theta_j}, \; \forall S \subseteq [K], \quad (2)$$

denotes $S^* := \operatorname{argmax}_{S \subseteq [K] | |S| \leq m} \mathcal{R}(S, \boldsymbol{\theta})$ is the optimal utility-maximizing subset. This objective corresponds to the standard objective in the MNL litterature Agrawal et al. (2019).

## 3 A Practical and Efficient Algorithm for AOA with PL

### 3.1 Algorithm Design

**Main Idea.** The crux of our novelty lies in our PL parameter estimation technique which maintains an estimate of pairwise scores of $p_{ij} = \frac{\theta_i}{\theta_i + \theta_j}$ for each pair of item $(i, j)$ using *Rank-Breaking* (RB) Khetan & Oh (2016)—a classical technique of extracting pairwise comparisons from choice (or partial ranking) feedback by breaking each win-loss pair independently in the choice data. A formal description is given in App. A.2. More precisely, using *rank-breaking* we estimate the relative (pairwise) strength $\widehat{p}_{ij,t}$ of each item pair $(i, j)$ at round $t$, as explained in (3). Further, noting $\theta_i = p_{i0}/(1 - p_{i0})$ (as $\theta_0 = 1$), we use $\widehat{p}_{i0,t}$ to estimate the MNL score $\widehat{\theta}_{i,t}$ of the $i$-th item using the NC (0-th item) as the 'pivot' item to benchmark against. Next, we prove a crucial concentration result in Lemma 1 showing indeed $\widehat{\theta}_{i,t}$ is a 'sharp' estimate of $\theta_i$, which is then subsequently used to prove the final regret guarantees Theorem 3 and Theorem 4 respectively. Our proposed algorithm Alg. 1 is described below:

**Estimate upper-confidence-bounds $\theta_t^{\texttt{ucb}}$ from Pairwise Estimates.** At each time $t$, our algorithm (Alg. 1) maintains a pairwise preference matrix $\widehat{\mathbf{P}}_t \in [0, 1]^{n \times n}$, whose $(i, j)$-th entry $\widehat{p}_{ij,t}$ records the empirical probability of $i$ having beaten $j$ in a pairwise duel, and a

corresponding upper confidence bound $p_{ij,t}^{\text{ucb}}$. Let $[\tilde{K}] := [K] \cup \{0\}$. We define for each pair $(i, j) \in [\tilde{K}] \times [\tilde{K}]$,

$$p_{ij,t}^{\text{ucb}} := \widehat{p}_{ij,t} + \sqrt{\frac{2\widehat{p}_{ij,t}(1 - \widehat{p}_{ij,t})x}{n_{ij,t}}} + \frac{3x}{n_{ij,t}}, \qquad \text{where} \quad \widehat{p}_{ij,t} := \frac{w_{ij,t}}{n_{ij,t}}, \qquad (3)$$

where $x > 0$ is an input of Alg. 1 and $w_{ij,t} = \sum_{s=1}^{t-1} \mathbb{1}\{i_s = i, j \in S_s\}$ denotes the number of pairwise wins of item-$i$ over $j$ after *rank-breaking* and $n_{ij,t} = w_{ij,t} + w_{ji,t}$ being the total number of times $(i, j)$ has been *'rank-broken'* till time $t$ (details in App. A.2). Noting that $\theta_i = p_{i0}/(1 - p_{i0})$, the above UCB estimates $p_{ij,t}^{\text{ucb}}$ are further used to design UCB estimates of the PL parameters $\theta_i$ as follows

$$\theta_{i,t}^{\text{ucb}} = p_{i0,t}^{\text{ucb}}/(1 - p_{i0,t}^{\text{ucb}})_+, \qquad \text{where} \quad (\cdot)_+ := \max\{\cdot, 0\}.$$

**Optimistic Assortment Selection** The estimates $\theta_{i,t}^{\text{ucb}}$s are then used to select the set $S_t$, that maximizes the underlying objective. This optimization problem transforms into a static assortment optimization problem with upper confidence bounds $\theta_{i,t}^{\text{ucb}}$ as the parameters, and efficient solution methods for this case are available (see e.g., Avadhanula et al. (2016); Davis et al. (2013); Rusmevichientong et al. (2010)).

---

**Algorithm 1 AOA for PL model with RB (AOA-RB$_{\text{PL}}$)**

1: **input:** $x > 0$
2: **init:** $\tilde{K} \leftarrow K + 1$, $[\tilde{K}] = [K] \cup \{0\}$, $\mathbf{W}_1 \leftarrow [0]_{\tilde{K} \times \tilde{K}}$
3: **for** $t = 1, 2, 3, \ldots, T$ **do**
4:      Set $\mathbf{N}_t = \mathbf{W}_t + \mathbf{W}_t^\top$, and $\widehat{\mathbf{P}}_t = \frac{\mathbf{W}_t}{\mathbf{N}_t}$. Denote $\mathbf{N}_t = [n_{ij,t}]_{\tilde{K} \times \tilde{K}}$ and $\widehat{\mathbf{P}}_t = [\widehat{p}_{ij,t}]_{\tilde{K} \times \tilde{K}}$.
5:      Define for all $i$, $p_{ii,t}^{\text{ucb}} = \frac{1}{2}$ and for all $i, j \in [\tilde{K}], i \neq j$

$$p_{ij,t}^{\text{ucb}} = \widehat{p}_{ij,t} + \left(\frac{2\widehat{p}_{ij,t}(1 - \widehat{p}_{ij,t})x}{n_{ij,t}}\right)^{1/2} + \frac{3x}{n_{ij,t}}$$

6:      $\theta_{i,t}^{\text{ucb}} := p_{i0,t}^{\text{ucb}}/(1 - p_{i0,t}^{\text{ucb}})_+$
7:      $S_t \leftarrow \begin{cases} \text{Top-}m \text{ items from argsort}(\{\theta_{1,t}^{\text{ucb}}, \ldots, \theta_{K,t}^{\text{ucb}}\}), \\ \qquad\qquad \text{for Top-}m \text{ objective} \\ \text{argmax}_{S \subseteq [K] || S| \leq m} \mathcal{R}(S, \theta_t^{\text{ucb}}), \\ \qquad\qquad \text{for Wtd-Top-}m \text{ objective} \end{cases}$
8:      Play $S_t$
9:      Receive the winner $i_t \in [\tilde{K}]$ (drawn as per (1))
10:      Update: $\mathbf{W}_{t+1} = [w_{ij,t+1}]_{\tilde{K} \times \tilde{K}}$ s.t. $w_{i_t j, t+1} \leftarrow w_{i_t j, t} + 1 \quad \forall j \in S_t \cup \{0\}$
11: **end for**

---

### 3.2 ANALYSIS: CONCENTRATION LEMMAS

We start the analysis by providing two technical lemmas, whose proofs are deferred to the appendix and that provide confidence bounds for the $\theta_i$.

**Lemma 1.** *Let $T \geq 1$ and $x > 0$. Then, with probability at least $1 - 3KTe^{-x}$, for all $t \in [T]$ and $i \in [K]$: $\theta_i \leq \theta_{i,t}^{ucb}$ atleast one of the following two inequalities is satisfied*

$$n_{i0,t} < 69x(\theta_0 + \theta_i) \quad or \quad \theta_{i,t}^{ucb} \leq \theta_i + 4(\theta_0 + \theta_i)\sqrt{\frac{2\theta_0\theta_i x}{n_{i0,t}}} + \frac{22x(\theta_0 + \theta_i)^2}{n_{i0,t}}.$$

The above lemma depends on $n_{i0,t}$ the number of times items $i$ have been compared with item 0 up to round $t$. The latter is controlled using the following lemma:

**Lemma 2.** *Let $T \geq 1$ and $x > 0$. Then, with probability at least $1 - KTe^{-x}$: simultaneously for all $t \in [T]$ and $i \in [K]$*

$$\tau_{i,t} < 2x(\theta_0 + \Theta_{S^*})^2 \quad or \quad n_{i0,t} \geq \frac{(\theta_0 + \theta_i)\tau_{i,t}}{2(\theta_0 + \Theta_{S^*})}, \qquad (4)$$

*where $\tau_{i,t} = \sum_{s=1}^{t-1} \mathbb{1}\{i \in S_s\}$ denotes the number of rounds item $i$ got selected before round $t$.*

### 3.3 Analysis: Top-$m$ Objective:

We are now ready to provide the regret upper bound for Algorithm 1 with Top-$m$ objective.

**Theorem 3** (Top-$m$ Objective). *Let $\theta_{\max} \geq 1$. Consider any instance of PL model on $K$ items with parameters $\theta \in [0, \theta_{\max}]^K$, $\theta_0 = 1$. The regret of Alg. 1 with parameter $x = 2 \log T$ is bounded as*

$$Reg_T^{top} = O\big(\theta_{\max}^{3/2} \sqrt{KT \log T}\big) \quad when \ T \to \infty.$$

The above rate of $\tilde{O}(\sqrt{KT})$ is optimal (up to log-factors), as a lower bound can be derived from standard multi-armed bandits Auer (2000); Auer et al. (2002). We only state here a sketch of the proof of Theorem 3. The detailed proof is deferred to the App. B.

*Proof Sketch of Theorem 3.* Let us define for any $S \subseteq [K]$,

$$\Theta_S = \sum_{i \in S} \theta_i, \quad and \quad \Theta_S^{\mathtt{ucb}} := \sum_{i \in S} \theta_i^{\mathtt{ucb}}.$$

Let $\mathcal{E}$ be the high-probability event such that both Lemma 1 and 2 holds true. Then, $\mathbb{P}(\mathcal{E}) \geq 1 - 4TKe^{-x}$. Let us first assume that $\mathcal{E}$ holds true. Then, by Lemma 1, $\Theta_{S^*} \leq \Theta_{S^*}^{\mathtt{ucb}} \leq \Theta_{S_t}^{\mathtt{ucb}}$, which yields

$$Reg_T^{\mathtt{top}} = \frac{1}{m} \sum_{t=1}^{T} \Theta_{S^*} - \Theta_{S_t} \leq \frac{1}{m} \sum_{t=1}^{T} \Theta_{S_t}^{\mathtt{ucb}} - \Theta_{S_t} \lesssim \tau_0 + \frac{1}{m} \sum_{t=1}^{T} \sum_{i \in S_t} (\theta_{i,t}^{\mathtt{ucb}} - \theta_i) \mathbb{1}\big\{\tau_{i,t} \geq \tau_0\big\},$$

where $\tau_0 = 138x(m+1)^2 \theta_{\max}^2$ corresponds to an exploration phase needed for the confidence upper bounds of Lem 1 and 2 to be satisfied. Then, noting that if $\mathcal{E}$ holds true, we can show by Lemma 2, that $\mathbb{1}\{\tau_{i,t} \geq \tau_0\} \leq \mathbb{1}\{n_{i0,t} \geq 69x(\theta_0 + \theta_i)\}$. Therefore, we can apply Lemma 1 that entails,

$$\frac{1}{m} \sum_{t=1}^{T} \sum_{i \in S_t} (\theta_{i,t}^{\mathtt{ucb}} - \theta_i) \mathbb{1}\big\{\tau_{i,t} \geq \bar{n}_{i0}\big\} \lesssim \frac{1}{m} \sum_{t=1}^{T} \sum_{i \in S_t} \left( (\theta_0 + \theta_i) \sqrt{\frac{\theta_0 \theta_i x}{n_{i0,t}}} \mathbb{1}\big\{\tau_{i,t} \geq \tau_0\big\} \right)$$

$$\overset{\text{Lem. 2}}{\lesssim} \frac{1}{m} \sum_{t=1}^{T} \sum_{i \in S_t} \theta_{\max}^{3/2} \sqrt{\frac{mx}{\tau_{i,t}}} \lesssim \frac{1}{m} \sum_{i=1}^{K} \theta_{\max}^{3/2} \sqrt{mx\tau_{i,t}} \lesssim \theta_{\max}^{3/2} \sqrt{xKT} \,.$$

where we used $\sum_{i=1}^{n} 1/\sqrt{i} \leq 2\sqrt{n}$ and $\sum_i \tau_{i,t} = mT$ together with Jensen's inequality in the last inequality. We thus have under the event $\mathcal{E}$ that $Reg_T^{\mathtt{top}} \leq O(\theta_{\max}^{3/2} \sqrt{xKT})$ and the proof is concluded by taking the expectation with $x = 2 \log T$ to control $\mathbb{P}(\mathcal{E}^c)$. □

### 3.4 Analysis: Wtd-Top-$m$ Objective

In this section we analyze the regret guarantee of Alg. 1 for Wtd-Top-$m$ objective (2).

**Theorem 4** (Wtd-Top-$m$ Objective). *Let $\theta_{\max} \geq 1$. Then, for any $\theta \in [0, \theta_{\max}]^K$ and weights $\mathbf{r} \in [0, 1]^K$, the weighted regret of AOA-RB$_{PL}$ (Alg. 1) with $x = 2 \log T$*

$$Reg_T^{wtd} = O(\sqrt{\theta_{\max} KT} \log T) \qquad when \quad T \to \infty.$$

The complete proof is postponed to App. B. The rate $\Omega(\sqrt{KT})$ is optimal as proved by the lower bound in Chen & Wang (2017) for MNL bandit problems for $\theta_{\max} = 1$. Our result recovers (up to a factor $\sqrt{\log T}$) the one of Agrawal et al. (2019) when $\theta_{\max} = 1$. However, their algorithm relies on more sophisticated estimators that necessitate epochs repeating the same assortment until the No-Choice item is selected. Note for our problem setting, where it is possible to have $\theta_{\max} \gg \theta_0 = 1$, the length of these epochs could be of $O(K\theta_{\max})$, which could be potentially very large when $\theta_{\max} \gg 1$. This reduces the number of effective epochs, leading to poor estimation of the PL parameters. We see this tradeoff in our experiments (Sec. 5) where the MNL-UCB algorithm of Agrawal et al. (2019) yields linear $O(T)$ regret for such choice of the problem parameters.

*Proof sketch of Thm. 4.* Let $\mathcal{E}$ be the high-probability event such that both Lemma 1 and 2 are satisfied. Then,

$$Reg_T^{\text{wtd}} = \sum_{t=1}^{T} \mathbb{E}\big[\mathcal{R}(S^*,\theta) - \mathcal{R}(S_t,\theta)\big] \lesssim \sum_{t=1}^{T} \mathbb{E}\big[(\mathcal{R}(S^*,\theta) - \mathcal{R}(S_t,\theta))\mathbb{1}\{\mathcal{E}\}\big] + T\mathbb{P}(\mathcal{E}^c)$$

$$\lesssim \sum_{t=1}^{T} \mathbb{E}\big[(\mathcal{R}(S_t,\theta_t^{\text{ucb}}) - \mathcal{R}(S_t,\theta))\mathbb{1}\{\mathcal{E}\}\big] + T\mathbb{P}(\mathcal{E}^c) \tag{5}$$

because $\mathcal{R}(S_t,\theta_t^{\text{ucb}}) \geq \mathcal{R}(S^*,\theta_t^{\text{ucb}}) \geq \mathcal{R}(S^*,\theta)$ under the event $\mathcal{E}$ by Lemma 8. We now upper-bound the first term of the right-hand-side

$$\sum_{t=1}^{T} \mathbb{E}\Big[\big((\mathcal{R}(S_t,\theta_t^{\text{ucb}}) - \mathcal{R}(S_t,\theta))\big)\mathbb{1}\{\mathcal{E}\}\Big] = \sum_{t=1}^{T} \mathbb{E}\Big[\Big(\sum_{i\in S_t} \frac{r_i\theta_{i,t}^{\text{ucb}}}{\theta_0 + \Theta_{S_t,t}^{\text{ucb}}} - \frac{r_i\theta_i}{\theta_0 + \Theta_{S_t}}\Big)\mathbb{1}\{\mathcal{E}\}\Big]$$

$$\leq \sum_{t=1}^{T} \mathbb{E}\Big[\Big(\sum_{i\in S_t} \frac{r_i(\theta_{i,t}^{\text{ucb}} - \theta_i)}{\theta_0 + \Theta_{S_t}}\Big)\mathbb{1}\{\mathcal{E}\}$$

Because $\Theta_{S_t,t}^{\text{ucb}} \geq \Theta_{S_t}$ under the event $\mathcal{E}$ by Lemma 1. Then, using $r_i \leq 1$, we further upper-bound using an exploration parameter $\tau_0 = O(\log(T))$ so that the upper-confidence-bounds in Lemmas 1 and 2 are satisfied

$$\sum_{t=1}^{T} \mathbb{E}\Big[\big((\mathcal{R}(S_t,\theta_t^{\text{ucb}}) - \mathcal{R}(S_t,\theta))\big)\mathbb{1}\{\mathcal{E}\}\Big] \leq \sum_{i=1}^{K} \mathbb{E}\Big[\sum_{t=1}^{T}\Big(\frac{|\theta_{i,t}^{\text{ucb}} - \theta_i|}{\theta_0 + \Theta_{S_t}}\Big)\mathbb{1}\{i \in S_t, \mathcal{E}\}\Big]$$

$$\lesssim O(\tau_0) + \sum_{i=1}^{K} \mathbb{E}\Big[\sum_{t=1}^{T} \frac{|\theta_{i,t}^{\text{ucb}} - \theta_i|}{\theta_0 + \Theta_{S_t}}\mathbb{1}\{i \in S_t, \tau_{i,t} \geq \tau_0, \mathcal{E}\}\Big]$$

$$\lesssim O(\tau_0) + \sum_{i=1}^{K} \sqrt{\sum_{t=1}^{T}\mathbb{E}\Big[\frac{\theta_i\mathbb{1}\{i \in S_t\}}{\theta_0 + \Theta_{S_t}}\Big]} \times \underbrace{\sqrt{\sum_{t=1}^{T}\mathbb{E}\Big[\Big(\frac{\theta_{i,t}^{\text{ucb}} - \theta_i}{\theta_0 + \Theta_{S_t}}\Big)^2 \frac{\theta_0 + \Theta_{S_t}}{\theta_i}\mathbb{1}\{i \in S_t, \tau_{i,t} \geq \tau_0, \mathcal{E}\}\Big]}}_{=:A_T(i)} \tag{6}$$

where the last inequality is by Cauchy-Schwarz inequality. Now, the term $A_T(i)$ above may be upper-bounded using Lemmas 1 and 2,

$$A_T(i) = \mathbb{E}\Big[\frac{(\theta_{i,t}^{\text{ucb}} - \theta_i)^2}{\theta_i(\theta_0 + \Theta_{S_t})}\mathbb{1}\{i \in S_t, \tau_{i,t} \geq \tau_0, \mathcal{E}\}\Big] \lesssim \sum_{t=1}^{T}\mathbb{E}\Big[\frac{(\theta_0 + \theta_i)^2 x}{n_{i0,t}(\theta_0 + \Theta_{S_t})}\mathbb{1}\{i \in S_t\}\Big]$$

$$\lesssim \theta_{\max} x \sum_{t=1}^{T}\mathbb{E}\Big[\frac{(\theta_0 + \theta_i)\mathbb{1}\{i \in S_t\}}{(\theta_0 + \Theta_{S_t})n_{i0,t}}\Big] = \theta_{\max} x \mathbb{E}\Big[\sum_{t=1}^{T}\frac{\mathbb{1}\{i_t \in \{i,0\}, i \in S_t\}}{n_{i0,t}}\Big] \lesssim \theta_{\max} x \log T$$

where in the last inequality we used that $\sum_{n=1}^{T} n^{-1} \leq 1 + \log T$. Substituting into (6), Jensen's inequality entails,

$$\sum_{t=1}^{T}\mathbb{E}\Big[(\mathcal{R}(S_t,\theta_t^{\text{ucb}}) - \mathcal{R}(S_t,\theta))\mathbb{1}\{\mathcal{E}\}\Big] \lesssim O(\tau_0) + \mathbb{E}\Big[\sqrt{\theta_{\max} x \log T}\sum_{i=1}^{K}\sqrt{\sum_{t=1}^{T}\frac{\theta_i\mathbb{1}\{i \in S_t\}}{\theta_0 + \Theta_{S_t}}}\Big]. \tag{7}$$

The proof is finally concluded by applying Cauchy-Schwarz inequality which yields:

$$\sum_{i=1}^{K}\sqrt{\sum_{t=1}^{T}\frac{\theta_i\mathbb{1}\{i \in S_t\}}{\theta_0 + \Theta_{S_t}}} \leq \sqrt{K\sum_{t=1}^{T}\frac{\sum_{i=1}^{K}\theta_i\mathbb{1}\{i \in S_t\}}{\theta_0 + \Theta_{S_t}}} \leq \sqrt{KT}.$$

Finally, combining the above result with (5) and (7) concludes the proof

$$Reg_T^{\text{wtd}} \lesssim TP(\mathcal{E}^c) + O(\tau_0) + \sqrt{\theta_{\max} x K T \log T}.$$

Choosing $x = 2\log T$ ensures $TP(\mathcal{E}^c) \leq O(1)$ and $\tau_0 \leq O(\log T)$. $\qquad\square$

## 4 Improved Dependance on $\theta_{\max}$ with Adaptive Pivot Selection

A problem with Algorithm 1 stems from estimating all $\theta_i$ based on pairwise comparisons with item 0. When $\theta_{\max} \gg \theta_0 = 1$, item 0 may not be sampled enough as the winner, leading to poor estimators. This deficiency contributes to the suboptimal dependence on $\theta_{\max}$ observed in Theorems 3 and 4 and in prior work, such as Agrawal et al. (2019). We propose the following fix to optimize the pivot. For all $i, j \in [K] \cup \{0\}$ we define $\gamma_{ij} = \frac{\theta_i}{\theta_j}$,

$$\gamma_{ij,t}^{\texttt{ucb}} = p_{ij,t}^{\texttt{ucb}} / (1 - p_{ij,t}^{\texttt{ucb}})_+ \qquad \text{and} \qquad \gamma_{ii,t}^{\texttt{ucb}} = 1\,,$$

where $p_{ij,t}^{\texttt{ucb}}$ are defined in (3). For all rounds $t$, the algorithm AOA-RB$_{\text{PL}}$-Adaptive selects

$$S_t = \underset{|S| \leq m}{\text{argmax}}\, \mathcal{R}(S, \widehat{\theta}_t^{\texttt{ucb}}) \qquad \text{where} \qquad \widehat{\theta}_{i,t}^{\texttt{ucb}} := \min_{j \in [K] \cup \{0\}} \gamma_{ij,t}^{\texttt{ucb}} \gamma_{j0,t}^{\texttt{ucb}}\,.$$

With the above definition of $\widehat{\theta}_{i,t}^{\texttt{ucb}}$, any item $i$ is compared to the base item 0 through the best possible item $j$. When $j$ is a strong item that is often selected both $\gamma_{ij,t}^{\texttt{ucb}}$ and $\gamma_{j0,t}^{\texttt{ucb}}$ are sharp upper-bounds of $\gamma_{ij}$ and $\gamma_{j0}$, making $\widehat{\theta}_{i,t}^{\texttt{ucb}}$ itself a sharp upper-confidence bound for $\theta_i$. This definition in turn also satisfies the condition $\widehat{\theta}_{i,t}^{\texttt{ucb}} \geq \theta_i$ required by Lemma 8 and crucial for our analysis. The condition would not hold if we only used $\gamma_{ij,t}^{\texttt{ucb}}$ without the $\gamma_{j0,t}^{\texttt{ucb}}$ factor.

We offer below a regret bound that underscores the value of optimizing the pivot when $\theta_{\max} \gg K$. Note that while the algorithm and analysis are presented for the weighted objective with winner feedback only, it can be adapted to other objectives by replacing $\mathcal{R}(S, \theta)$ with the new objective in the analysis, as long as Lemma 8 remains valid.

**Theorem 5.** *Let $\theta_{\max} \geq 1$. For any $\theta \in [0, \theta_{\max}]^K$ and weights $\mathbf{r} \in [0, 1]^K$, the weighted regret of AOA-RB$_{PL}$-Adaptive is upper-bounded as*

$$Reg_T^{wtd} = O\big(\sqrt{\min\{\theta_{\max}, K\} KT} \log T\big)$$

*as $T \to \infty$ for the choice $x = 2 \log T$ (when definining $p_{ij,t}^{ucb}$).*

The proof of Theorem 5 is deferred to the App. B, with a key step relying on selecting the pivot $j_t = \text{argmax}_{j \in S_t \cup \{0\}} \theta_j$. The use of $|\widehat{\theta}_{i,t}^{\texttt{ucb}} - \theta_i| \leq |\gamma_{ij_t,t}^{\texttt{ucb}} - \theta_i|$ provides confidence upper-bounds with an improved dependence on $\theta_{\max}$, leveraging the fact that $\theta_{j_t} \geq \theta_i$. Due to the varying pivot over time, a telescoping argument introduces an additive factor $\sqrt{K}$.

When $\theta_{\max}$ is constant, the regret is $O(K\sqrt{T} \log T)$, eliminating any asymptotic dependence on $\theta_{\max}$. This allows for handling scenarios where the No-Choice item is highly unlikely, unlike previous works such as Agrawal et al. (2019; 2017). Agrawal et al. (2019) did attempt to relax the assumption of $\theta_{\max} = \theta_0$ and shows a bound of order $O\big(\max\{\theta_{\max}/\theta_0, 1\}^{1/2} \sqrt{KT}\big)$, which unfortunately blows to $\infty$ as $\theta_0 \to 0$ (equiv. $\theta_{\max} \to \infty$).

**Remark 1** (Beyond MNL Assortment: Extending to any general RUM based Choice Models). *Although, in this paper, we primarily focused on MNL based choice models, our proposed algorithms may be generalized to more general random utility-based models (RUMs) Azari et al. (2012b); Saha & Ghoshal (2022) pursuing the ideas from Saha & Gopalan (2020) that extends the RB based parameter estimation technique to any RUM($\boldsymbol{\theta}$) choice models: Precisely, using the RB based RUM-parameter estimation technique of Saha & Gopalan (2020), we can show a regret bound of $\tilde{O}(c_{rum}^{-1} \sqrt{\min\{\theta_{\max}, K\} KT})$ for our proposed algorithm AOA-RBPL, where $c_{rum}$ is the parameter associated to the minimum advantage ratio (min-AR) of the underlying RUM($\boldsymbol{\theta}$) model, as defined in Thm. 6 of Saha & Gopalan (2020). In particular, $c_{rum}$ can shown to be a constant given a fixed RUM model, e.g. $c_{rum} = 1/4$ for Exp(1), Gamma(2, 1), $c_{rum} = 1/(4\sigma)$ for Gumbel($\mu, \sigma$), $c_{rum} = \lambda/4$ for Weibull($\lambda, 1$), $c_{rum} = 1/3$ for Gaussian(0, 1), etc (using Cor. 5 of Saha & Gopalan (2020)).*

## 5 Experiments

We run experiments to compare the performance of our method with the state-of-the-art methods. All results are averaged across 100 runs. We evaluate the performance of our main algorithm AOA-RB$_{\text{PL}}$-Adaptive (Sec. 4), referred as "Our Alg-1 (Adaptive Pivot)", with the

following algorithms: AOA-RB$_{\text{PL}}$ (Sec. 3) referred as "Our Alg-2 (No-Choice Pivot)", and MNL-UCB, the state-of-the-art algorithm for AOA (Agrawal et al. (2019), Alg. 1).

**Different PL ($\boldsymbol{\theta}$) Environments.** We report our experiment results on two datasets with $K = 50$ items: (1) Arith50 with PL parameters $\theta_i = 1 - (i-1)0.02$, $\forall i \in [50]$. (2) Bad50 with PL parameters $\theta_i = 0.6$, $\forall i \in [50] \setminus \{25\}$ and $\theta_{25} = 0.8$. For simplicity of computing the assortment choices $S_t$, we assume $r_i = 1$, $\forall i \in [K]$.

**(1). Averaged Regret with weak NC** ($\theta_{\max}/\theta_0 \gg 1$) **(Fig. 1):** In our first experiment, we set $m = 5$ and $\theta_0/\theta_{\max} = 0.01$ and report the average regret of the above three algorithms for our two objectives.

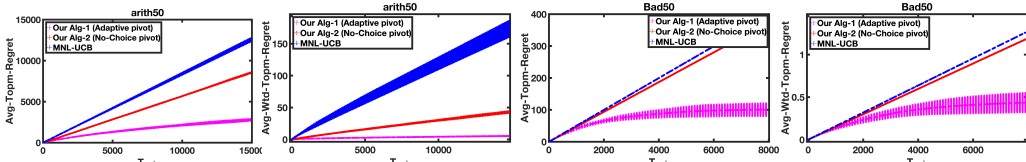

Figure 1: Averaged Regret for $m = 5$, $\theta_0 = 0.01$

Fig. 1 shows that our algorithm AOA-RB$_{\text{PL}}$-Adaptive (with adaptive pivot) significantly outperforms the other two algorithms, while our algorithm AOA-RB$_{\text{PL}}$ with no-choice (NC) pivot still outperforms MNL-UCB.

**(2). Averaged Regret vs No-Choice PL Parameter** ($\theta_{\max}/\theta_0$) **(Fig. 2):** In this experiment, we evaluate the regret performance of our algorithm AOA-RB$_{\text{PL}}$-Adaptive. We report the experiment on Artith50 PL dataset and set the subsetsize $m = 5$, $\theta_{\max}/\theta_0 = \{1, 0.5, 0.1, 0.05, 0.01, 0.005, 0.001\}$. Fig. 2 shows the increase in the performance gap between our algorithm AOA-RB$_{\text{PL}}$-Adaptive (with adaptive pivot) with decreasing $\theta_0/\theta_{\max}$.

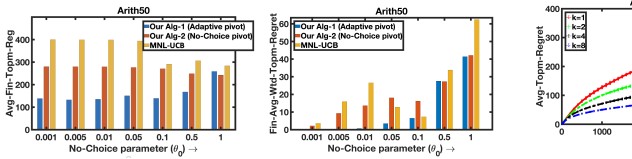

Figure 2: Comparative performance for varying $\theta_0/\theta_{\max}$, $m = 5$

Figure 3: Tradofff: Averaged Regret vs length of the $k$ rank-ordered feedback

**(3). Averaged Regret vs Length of the rank-ordered feedback** ($k$) **(Fig. 3):** We also run a thought experiment to understand the tradeoff between learning rate with $k$-length rank-ordered feedback, where given any assortment $S_t \subseteq [K]$ of size $m$, the learner gets to see the top-$k$ draws ($k \le m$) from the PL model without replacement. This is a stronger feedback than the winner (i.e. top-1 for $k = 1$) feedback and, as expected, we see in Fig. 3 an improved regret (for both notions) when increasing $k$. The experiment are run on the Artith50 dataset with $m = 30$ and $k \in \{1, 2, 4, 8\}$.

## 6 CONCLUSION

We study the Active Optimal Assortment Selection problem under PL choice models, introducing a framework (*AOA*). Our algorithm uses a novel 'Rank-Breaking' technique for tight parameter estimation, ensuring efficiency, optimality (up to log factors), and practicality without restrictive assumptions on the default (no-choice) item.

**Future Works.** Among many interesting questions to address in the future, it will be interesting to understand the role of the No-Choice (NC) item in the algorithm design — An intriguing direction is whether similar results can be achieved or improved using recent parameter estimation technique for MNL models such as the one from Agarwal et al. (2018) instead of rank-breaking. However, as currently formulated, their technique suffers from an undesirable dependence that scales with $(\theta_{\max}/\theta_0)^2$. This dependence might be mitigated in their analysis by employing batches to estimate the parameters $\theta_i$. However, it remains unclear whether their approach can completely eliminate the dependence on $\theta_{\max}$.

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

# SUPPLEMENTARY: FINALLY RANK-BREAKING CONQUERS MNL BANDITS: OPTIMAL AND EFFICIENT ALGORITHMS FOR MNL ASSORTMENT

## A PRELIMINARIES: SOME USEFUL CONCEPTS FOR PL CHOICE MODELS

### A.1 MNL: A DISCRETE CHOICE MODEL

A discrete choice model specifies the relative preferences of two or more discrete alternatives in a given set. A widely studied class of discrete choice models is the class of *Random Utility Models* (RUMs), which assume a ground-truth utility score $\theta_i \in \mathbb{R}$ for each alternative $i \in [n]$, and assign a conditional distribution $\mathcal{D}_i(\cdot|\theta_i)$ for scoring item $i$. To model a winning alternative given any set $S \subseteq [n]$, one first draws a random utility score $X_i \sim \mathcal{D}_i(\cdot|\theta_i)$ for each alternative in $S$, and selects an item with the highest random score.

One widely used RUM is the *Multinomial-Logit (MNL)* or *Plackett-Luce model (PL)*, where the $\mathcal{D}_i$s are taken to be independent Gumbel distributions with parameters $\theta_i'$ (Azari et al., 2012a), i.e., with probability densities

$$\mathcal{D}_i(x_i|\theta_i') = e^{-(x_j - \theta_j')} e^{-e^{-(x_j - \theta_j')}}, \qquad \theta_i' \in R, \ \forall i \in [n].$$

Moreover assuming $\theta_i' = \ln \theta_i$, $\theta_i > 0 \ \forall i \in [n]$, it can be shown in this case the probability that an alternative $i$ emerges as the winner in the set $S \ni i$ becomes: $\mathbb{P}(i|S) = \frac{\theta_i}{\sum_{j \in S} \theta_j}$.

Other families of discrete choice models can be obtained by imposing different probability distributions over the utility scores $X_i$, e.g. if $(X_1, \dots X_n) \sim \mathcal{N}(\boldsymbol{\theta}, \boldsymbol{\Lambda})$ are jointly normal with mean $\boldsymbol{\theta} = (\theta_1, \dots \theta_n)$ and covariance $\boldsymbol{\Lambda} \in \mathbb{R}^{n \times n}$, then the corresponding RUM-based choice model reduces to the *Multinomial Probit (MNP)*.

### A.2 RANK BREAKING

*Rank breaking* (RB) is a well-understood idea involving the extraction of pairwise comparisons from (partial) ranking data, and then building pairwise estimators on the obtained pairs by treating each comparison independently (Khetan & Oh, 2016; Jang et al., 2017), e.g., a winner $a$ sampled from among $a, b, c$ is rank-broken into the pairwise preferences $a \succ b$, $a \succ c$. We use this idea to devise estimators for the pairwise win probabilities $p_{ij} = \mathbb{P}(i|\{i, j\}) = \theta_i/(\theta_i + \theta_j)$ for our problem setting. We used the idea of RB in both our algorithms (AOA-RB$_{PL}$ and AOA-RB$_{PL}$-Adaptive) to update the pairwise win-count estimates $w_{i,j,t}$ for all the item pairs $(i, j) \in [K] \times [K]$, which is further used for deriving the empirical pairwise preference estimates $\widehat{p}_{ij,t}$, at any time $t$.

### A.3 PARAMETER ESTIMATION WITH PL BASED PREFERENCE DATA

**Lemma 6** (Pairwise win-probability estimates for the PL model (Saha & Gopalan, 2018)). *Consider a MNL model with parameters $\boldsymbol{\theta} = (\theta_1, \theta_2, \dots, \theta_n)$, and fix two items $i, j \in [n]$. Let $S_1, \dots, S_T$ be a sequence of (possibly random) subsets of $[n]$ of size at least 2, where $T$ is a positive integer, and $i_1, \dots, i_T$ a sequence of random items with each $i_t \in S_t$, $1 \le t \le T$, such that for each $1 \le t \le T$, (a) $S_t$ depends only on $S_1, \dots, S_{t-1}$, and (b) $i_t$ is distributed as the MNL winner of the subset $S_t$, given $S_1, i_1, \dots, S_{t-1}, i_{t-1}$ and $S_t$, and (c) $\forall t : \{i, j\} \subseteq S_t$ with probability 1. Let $n_i(T) = \sum_{t=1}^{T} \mathbb{P}(i_t = i)$ and $n_{ij}(T) = \sum_{t=1}^{T} \mathbb{P}(\{i_t \in \{i, j\}\})$. Then, for any positive integer $v$, and $\eta \in (0, 1)$,*

$$\mathbb{P}\left( \frac{n_i(T)}{n_{ij}(T)} - \frac{\theta_i}{\theta_i + \theta_j} \ge \eta, \ n_{ij}(T) \ge v \right) \le e^{-2v\eta^2},$$

$$\mathbb{P}\left( \frac{n_i(T)}{n_{ij}(T)} - \frac{\theta_i}{\theta_i + \theta_j} \le -\eta, \ n_{ij}(T) \ge v \right) \le e^{-2v\eta^2}.$$

# B    OMITTED PROOFS FROM SEC. 3 AND SEC. 4

## B.1    A CONCENTRATION BOUNDS FOR THE $p_{ij,t}$

We first prove below a concentration inequality based on Bernstein's inequality for the estimators $p_{ij,t}$.

**Lemma 7.** *Let $(i,j) \in [K] \times [K]$. Let $T \geq 1$ and $x > 0$. Then, with probability at least $1 - 3Te^{-x}$,*

$$p_{ij} \leq p_{ij,t}^{\mathtt{ucb}} \leq p_{ij} + 2\sqrt{\frac{2p_{ij}(1-p_{ij})x}{n_{ij,t}}} + \frac{11x}{n_{ij,t}}, \tag{8}$$

*simultaneously for all $t \in [T]$.*

*Proof of Lemma 7.* Let $T \geq 1$, $x > 0$ and $i,j \in [K]$. Applying Thm. 1 of Audibert et al. (2009), with probability at least $1 - \beta(x,T)$, we get simultaneously for all $t \in [T]$,

$$\left|\widehat{p}_{ij,t} - p_{ij}\right| \leq \sqrt{\frac{2\widehat{p}_{ij,t}(1-\widehat{p}_{ij,t})x}{n_{ij,t}}} + \frac{3x}{n_{ij,t}}, \tag{9}$$

where $\beta(x,T) = 3\inf_{1<\alpha\leq 3}\min\left\{\frac{\log T}{\log \alpha}, T\right\}e^{-x/\alpha} \leq 3Te^{-x}$. Note that the inequality holds true although $n_{ij,t}$ is a random variable. This, shows the first inequality

$$p_{ij} \leq p_{ij,t}^{\mathtt{ucb}}.$$

For the second inequality, (9) implies

$$p_{ij,t}^{\mathtt{ucb}} = \widehat{p}_{ij,t} + \sqrt{\frac{2\widehat{p}_{ij,t}(1-\widehat{p}_{ij,t})x}{n_{ij,t}}} + \frac{3x}{n_{ij,t}}$$

$$\leq p_{ij} + 2\sqrt{\frac{2\widehat{p}_{ij,t}(1-\widehat{p}_{ij,t})x}{n_{ij,t}}} + \frac{6x}{n_{ij,t}}. \tag{10}$$

Furthermore, because $x \mapsto x(1-x)$ is 1-Lipschitz on $[0,1]$, we have

$$\left|\widehat{p}_{ij,t}(1-\widehat{p}_{ij,t}) - p_{ij}(1-p_{ij})\right| \leq \left|\widehat{p}_{ij,t} - p_{ij}\right|$$

$$\overset{(9)}{\leq} \sqrt{\frac{2\widehat{p}_{ij,t}(1-\widehat{p}_{ij,t})x}{n_{ij,t}}} + \frac{3x}{n_{ij,t}}.$$

Therefore,

$$\widehat{p}_{ij,t}(1-\widehat{p}_{ij,t}) \leq p_{ij}(1-p_{ij}) + \sqrt{\frac{2\widehat{p}_{ij,t}(1-\widehat{p}_{ij,t})x}{n_{ij,t}}} + \frac{3x}{n_{ij,t}}$$

$$\leq \left(\sqrt{p_{ij}(1-p_{ij})} + \sqrt{\frac{3x}{n_{ij,t}}}\right)^2,$$

which yields

$$\sqrt{\widehat{p}_{ij,t}(1-\widehat{p}_{ij,t})} \leq \sqrt{p_{ij}(1-p_{ij})} + \sqrt{\frac{3x}{n_{ij,t}}}. \tag{11}$$

Plugging back into (10), we get

$$p_{ij,t}^{\mathtt{ucb}} \leq 2\sqrt{\frac{2p_{ij}(1-p_{ij})x}{n_{ij,t}}} + \frac{11x}{n_{ij,t}}.$$

$\square$

### B.2 Proof of Lemma 1

*Proof.* Let $i \in [K]$ and $x > 0$. Then, by a union bound on Lemma 7 and 2, with probability at least $1 - 4Te^{-x}$, (8) and (4) hold true for all $t \in [T]$. We consider this high-probability event in the rest of the proof. Define the function $f : x \mapsto x/(1-x)_+$ on $[0,1]$ (with the convention $f(1) = +\infty$), so that $\theta_{i,t}^{\mathrm{ucb}} = f(p_{i0,t}^{\mathrm{ucb}})$ and $\theta_i = f(p_{i0})$. Because $f$ is non-decreasing, and $p_{i0,t}^{\mathrm{ucb}} \geq p_{i0}$ by (8), we have

$$\theta_{i,t}^{\mathrm{ucb}} \geq \theta_i \,. \tag{12}$$

Furthermore, denote

$$\Delta_{i,t} := 2\sqrt{\frac{2p_{ij}(1-p_{ij})x}{n_{i0,t}}} + \frac{11x}{n_{i0,t}} = 2\sqrt{\frac{2\theta_0\theta_i x}{(\theta_0 + \theta_i)^2 n_{i0,t}}} + \frac{11x}{n_{i0,t}} \,. \tag{13}$$

In the rest of the proof we assume, $n_{i0,t} \geq 69x(\theta_0 + \theta_i)$. Then, using that $\theta_0\theta_i \leq \theta_0 + \theta_i$ since $\theta_0 = 1$, it implies

$$(\theta_0 + \theta_i)\Delta_{i,t} \leq 2\sqrt{\frac{2\theta_0\theta_i x}{n_{i0,t}}} + \frac{11x(\theta_0 + \theta_i)}{n_{i0,t}} \leq \frac{1}{2}\,,$$

and

$$p_{i0} + \Delta_{i,t} = \frac{\theta_i}{\theta_0 + \theta_i} + \Delta_{i,t} \leq \frac{\theta_i + 1/2}{\theta_i + 1} < 1.$$

Thus, because $f$ is non-decreasing

$$\begin{aligned}
\theta_{i,t}^{\mathrm{ucb}} - \theta_i &= f(p_{i0,t}^{\mathrm{ucb}}) - f(p_{i0}) \\
&\overset{(8)}{\leq} f\big(p_{i0} + \Delta_{i,t}\big) - f(p_{i0}) \\
&= \frac{p_{i0} + \Delta_{i,t}}{1 - p_{i0} - \Delta_{i,t}} - \frac{p_{i0}}{1 - p_{i0}} \\
&= \frac{\Delta_{i,t}}{(1 - p_{i0})(1 - p_{i0} - \Delta_{i,t})} \\
&= \frac{(\theta_0 + \theta_i)^2 \Delta_{i,t}}{1 - (\theta_0 + \theta_i)\Delta_{i,t}} \\
&\leq 2(\theta_0 + \theta_i)^2 \Delta_{i,t} \\
&\overset{(13)}{\leq} 4(\theta_0 + \theta_i)\sqrt{\frac{2\theta_0\theta_i x}{n_{i0,t}}} + \frac{22x(\theta_0 + \theta_i)^2}{n_{i0,t}} \,,
\end{aligned}$$

which concludes the proof. $\qquad\square$

### B.3 Proof of Lemma 2

*Proof.* Let $T \geq 1$ and $i \in [K]$. Recall that $\tau_{i,t} = \sum_{s=1}^{t-1} \mathbb{1}\{i \in S_s\}$ is the number of times $i$ was played at the start of round $t$ and $n_{i0,t} = \sum_{s=1}^{t-1} \mathbb{1}\{i_t \in \{i, 0\}, i \in S_t\}$ is the number of times $i$ or 0 won up to round $t$ when played together. When $i$ is played the probability of 0 or $i$ to win is

$$\mathbb{P}(i_t \in \{i, 0\}|S_t) = \frac{\theta_0 + \theta_i}{\theta_0 + \Theta_{S_t}} \geq \frac{\theta_0 + \theta_i}{\theta_0 + \Theta_{S^*}} \,.$$

Therefore, applying Chernoff-Hoeffding inequality together with a union bound (to deal with the fact that $\tau_{i,t}$ is random), we have with probability at least $1 - Te^{-x}$

$$n_{i0,t} \geq \frac{\theta_0 + \theta_i}{\theta_0 + \Theta_{S^*}}\tau_{i,t} - \sqrt{\frac{\tau_{i,t}x}{2}}$$

simultaneously for all $t \in [T]$. Noting that

$$\frac{\theta_0 + \theta_i}{\theta_0 + \Theta_{S^*}}\tau_{i,t} - \sqrt{\frac{\tau_{i,t}x}{2}} \geq \frac{\theta_0 + \theta_i}{2(\theta_0 + \Theta_{S^*})}\tau_{i,t}$$

if $\tau_{i,t} \geq 2x(\theta_0 + \Theta_{S^*})^2 \geq \frac{2x(\theta_0 + \Theta_{S^*})^2}{(\theta_0 + \theta_i)^2}$ concludes the proof. $\qquad\square$

### B.4 Proof of Theorem 3

*Proof.* Let us define for any $S \subseteq [K]$,

$$\Theta_S = \sum_{i \in S} \theta_i, \quad \text{and} \quad \Theta_S^{\texttt{ucb}} := \sum_{i \in S} \theta_i^{\texttt{ucb}}.$$

Let $\mathcal{E}$ be the high-probabality event such that both Lemma 1 and 2 holds true. Then, $\mathbb{P}(\mathcal{E}) \geq 1 - 4TKe^{-x}$. Let us first assume that $\mathcal{E}$ holds true. Then, by Lemma 1,

$$Reg_T^{\texttt{top}} = \frac{1}{m} \sum_{t=1}^T \Theta_{S^*} - \Theta_{S_t}$$

$$\leq \frac{1}{m} \sum_{t=1}^T \min \left\{ \Theta_{S^*}, \Theta_{S_t}^{\texttt{ucb}} - \Theta_{S_t} \right\} \quad \leftarrow \quad \text{because } \Theta_{S^*} \leq \Theta_{S^*}^{\texttt{ucb}} \leq \Theta_{S_t}^{\texttt{ucb}} \text{ under the event } \mathcal{E}$$

$$= \frac{1}{m} \sum_{t=1}^T \min \left\{ \Theta_{S^*}, \sum_{i \in S_t} \theta_{i,t}^{\texttt{ucb}} - \theta_i \right\}$$

$$\leq \frac{1}{m} \Theta_{S^*} \sum_{i=1}^K \bar{\tau}_{i0} + \frac{1}{m} \sum_{t=1}^T \sum_{i \in S_t} (\theta_{i,t}^{\texttt{ucb}} - \theta_i) \mathbb{1}\{\tau_{i,t} \geq \bar{\tau}_{i0}\}$$

where $\bar{\tau}_{i0} = 2x(\theta_0 + \Theta_{S^*}) \max\{\theta_0 + \Theta_{S^*}, 69\} \leq 138x(m+1)^2\theta_{\max}^2$, where $\theta_{\max} := \max_i \theta_i$. Then, noting that if $\mathcal{E}$ holds true, by Lemma 2, we also have $n_{i0,t} \geq \frac{1}{2(\theta_0 + \Theta_{S^*})}(\theta_0 + \theta_i)\tau_{i,t}$, which yields

$$\mathbb{1}\{\tau_{i,t} \geq \bar{\tau}_{i0}\} \leq \mathbb{1}\{n_{i0,t} \geq 69x(\theta_0 + \theta_i)\}.$$

Therefore, we can apply Lemma 1 that entails,

$$\frac{1}{m} \sum_{t=1}^T \sum_{i \in S_t} (\theta_{i,t}^{\texttt{ucb}} - \theta_i) \mathbb{1}\{\tau_{i,t} \geq \bar{\tau}_{i0}\}$$

$$\overset{\text{Lem. 1}}{\leq} \frac{1}{m} \sum_{t=1}^T \sum_{i \in S_t} \left( 4(\theta_0 + \theta_i)\sqrt{\frac{2\theta_0\theta_i x}{n_{i0,t}}} + \frac{22x(\theta_0 + \theta_i)^2}{n_{i0,t}} \right) \mathbb{1}\{n_{i0,t} \geq 69x(\theta_0 + \theta_i)\}$$

$$\overset{\text{Lem 2}}{\leq} \frac{1}{m} \sum_{t=1}^T \sum_{i \in S_t} \left( 8\sqrt{\frac{(\theta_0 + \Theta_{S^*})(\theta_0 + \theta_i)\theta_0\theta_i x}{\tau_{i,t}}} + \frac{44x(\theta_0 + \Theta_{S^*})(\theta_0 + \theta_i)}{\tau_{i,t}} \right)$$

$$\leq \frac{1}{m} \sum_{i=1}^K 16\sqrt{(\theta_0 + \Theta_{S^*})(\theta_0 + \theta_i)\theta_0\theta_i x\tau_{i,T}} + 44x(\theta_0 + \Theta_{S^*}) \sum_{i=1}^K (\theta_0 + \theta_i)(1 + \log(\tau_{i,T})),$$

where we used $\sum_{i=1}^n 1/\sqrt{i} \leq 2\sqrt{n}$ and $\sum_{i=1}^n i^{-1} \leq 1 + \log n$. We thus have

$$Reg_T^{\texttt{top}} \leq 138x(m+1)^2 K\theta_{\max}^3 + \frac{1}{m} \sum_{i=1}^K 16\theta_{\max}^{3/2}\sqrt{(m+1)x\tau_{i,T}}$$

$$+ 44x(m+1)(1 + \theta_{\max})^2 \sum_{i=1}^K (1 + \log(\tau_{i,T}))$$

$$\leq 138x(m+1)^2 K\theta_{\max}^3 + 16\theta_{\max}^{3/2}\sqrt{2xKT} + 88x(m+1)K\theta_{\max}^2 \left(1 + \log\left(\frac{mT}{K}\right)\right).$$

Therefore,

$$\mathbb{E}[Reg_T^{\texttt{top}}] \leq 12\sqrt{2}xmK\theta_{\max}^3 + 16\theta_{\max}^{3/2}\sqrt{2xKT} + 88xmK\theta_{\max}^2 \left(1 + \log\left(\frac{mT}{K}\right)\right)$$

$$+ 4mKT^2 e^{-x}\theta_{\max}.$$

Choosing $x = 2\log T$ concludes the proof. $\qquad\square$

### B.5 PROOF OF THEOREM 4

We start by noting a result that shows that the expected utility $\mathcal{R}(S^*, \theta)$ that corresponds to the optimal assortment $S^* = \mathrm{argmax}_{S \subset [K], |S| \leq m} \mathcal{R}(S, \theta)$ is non-decreasing in the parameters $\theta$.

**Lemma 8** (Lemma A.3 of Agrawal et al. (2019)). *Let* $S^* = \mathrm{argmax}_{S \subset [K], |S| \leq m} \mathcal{R}(S, \theta)$. *Assume* $\theta_i^{ucb} \geq \theta_i$ *for all* $i \in [K]$, *then* $\mathcal{R}(S^*, \theta) \leq \mathcal{R}(S^*, \theta^{ucb})$.

*Proof of Theorem 4.* Let $\mathcal{E}$ be the high-probability event such that Lemma 1 and 2 are satisfied, so that $\mathbb{P}(\mathcal{E}) \geq 1 - 4KTe^{-x}$. Then, denoting $x \wedge y := \min\{x, y\}$,

$$Reg_T^{\text{wtd}} = \sum_{t=1}^{T} \mathbb{E}\big[\mathcal{R}(S^*, \theta) - \mathcal{R}(S_t, \theta)\big] \tag{14}$$

$$= \sum_{t=1}^{T} \mathbb{E}\big[(\mathcal{R}(S^*, \theta) - \mathcal{R}(S_t, \theta))\mathbb{1}\{\mathcal{E}\} + (\mathcal{R}(S^*, \theta) - \mathcal{R}(S_t, \theta))\mathbb{1}\{\mathcal{E}^c\}\big]$$

$$\leq \sum_{t=1}^{T} \mathbb{E}\Big[((\mathcal{R}(S_t, \theta_t^{\text{ucb}}) - \mathcal{R}(S_t, \theta)) \wedge \mathcal{R}(S^*, \theta))\mathbb{1}\{\mathcal{E}\} + \mathcal{R}(S^*, \theta)\mathbb{1}\{\mathcal{E}^c\}\Big]$$

because $\mathcal{R}(S_t, \theta_t^{\text{ucb}}) \geq \mathcal{R}(S^*, \theta_t^{\text{ucb}}) \geq \mathcal{R}(S^*, \theta)$ under the event $\mathcal{E}$ by Lemma 8. Then, using $\mathcal{R}(S^*, \theta) \leq \max_i r_i \leq 1$, we get

$$Reg_T^{\text{wtd}} \leq \sum_{t=1}^{T} \mathbb{E}\Big[((\mathcal{R}(S_t, \theta_t^{\text{ucb}}) - \mathcal{R}(S_t, \theta)) \wedge 1)\mathbb{1}\{\mathcal{E}\} + \mathbb{1}\{\mathcal{E}^c\}\Big]$$

$$\leq 4T^2 K e^{-x} + \sum_{t=1}^{T} \mathbb{E}\Big[\Big((\mathcal{R}(S_t, \theta_t^{\text{ucb}}) - \mathcal{R}(S_t, \theta)) \wedge 1\Big)\mathbb{1}\{\mathcal{E}\}\Big].$$

Let us upper-bound the second term of the right-hand-side

$$\sum_{t=1}^{T} \mathbb{E}\Big[\Big((\mathcal{R}(S_t, \theta_t^{\text{ucb}}) - \mathcal{R}(S_t, \theta)) \wedge 1\Big)\mathbb{1}\{\mathcal{E}\}\Big] \tag{15}$$

$$= \sum_{t=1}^{T} \mathbb{E}\Big[\Big(\Big(\sum_{i \in S_t} \frac{r_i \theta_{i,t}^{\text{ucb}}}{\theta_0 + \Theta_{S_t,t}^{\text{ucb}}} - \frac{r_i \theta_i}{\theta_0 + \Theta_{S_t}}\Big) \wedge 1\Big)\mathbb{1}\{\mathcal{E}\}\Big]$$

$$\leq \sum_{t=1}^{T} \mathbb{E}\Big[\Big(\Big(\sum_{i \in S_t} \frac{r_i(\theta_{i,t}^{\text{ucb}} - \theta_i)}{\theta_0 + \Theta_{S_t}}\Big) \wedge 1\Big)\mathbb{1}\{\mathcal{E}\}\Big] \qquad \text{because } \Theta_{S_t,t}^{\text{ucb}} \geq \Theta_{S_t} \text{ under } \mathcal{E}$$

$$\leq \sum_{t=1}^{T} \mathbb{E}\Big[\Big(\Big(\sum_{i \in S_t} \frac{|\theta_{i,t}^{\text{ucb}} - \theta_i|}{\theta_0 + \Theta_{S_t}}\Big) \wedge 1\Big)\mathbb{1}\{\mathcal{E}\}\Big] \qquad \text{because } r_i \leq 1$$

$$\leq \sum_{i=1}^{K} \mathbb{E}\Big[\sum_{t=1}^{T} \Big(\frac{|\theta_{i,t}^{\text{ucb}} - \theta_i|}{\theta_0 + \Theta_{S_t}} \wedge 1\Big)\mathbb{1}\{i \in S_t\}\mathbb{1}\{\mathcal{E}\}\Big]$$

$$\leq 138xm^2 K\theta_{\max}^2 + \sum_{i=1}^{K} \mathbb{E}\Big[\sum_{t=1}^{T} \frac{|\theta_{i,t}^{\text{ucb}} - \theta_i|}{\theta_0 + \Theta_{S_t}}\mathbb{1}\{i \in S_t, \tau_{i,t} \geq 138x(m+1)^2\theta_{\max}^2\}\mathbb{1}\{\mathcal{E}\}\Big]$$

$$\leq 138xm^2 K\theta_{\max}^2 + \sum_{i=1}^{K} \sqrt{\sum_{t=1}^{T} \mathbb{E}\Big[\frac{(\frac{\theta_0}{m} + \theta_i)\mathbb{1}\{i \in S_t\}}{\theta_0 + \Theta_{S_t}}\Big]}$$

$$\times \underbrace{\sqrt{\sum_{t=1}^{T} \mathbb{E}\Big[\Big(\frac{|\theta_{i,t}^{\text{ucb}} - \theta_i|}{\theta_0 + \Theta_{S_t}}\Big)^2 \frac{\theta_0 + \Theta_{S_t}}{\frac{\theta_0}{m} + \theta_i}\mathbb{1}\{i \in S_t, \tau_{i,t} \geq 138x(m+1)^2\theta_{\max}^2\}\mathbb{1}\{\mathcal{E}\}\Big]}}_{=:A_T(i)} \tag{16}$$

where the last inequality is by Cauchy-Schwarz inequality. Now, the term $A_T(i)$ above may be upper-bounded as follows

$$A_T(i) := \sum_{t=1}^{T} \mathbb{E}\left[\left(\frac{|\theta_{i,t}^{\mathtt{ucb}} - \theta_i|}{\theta_0 + \Theta_{S_t}}\right)^2 \frac{\theta_0 + \Theta_{S_t}}{\frac{\theta_0}{m} + \theta_i} \mathbb{1}\{i \in S_t, \tau_{i,t} \geq 138x(m+1)^2\theta_{\max}^2\}\mathbb{1}\{\mathcal{E}\}\right]$$

$$= \mathbb{E}\left[\frac{(\theta_{i,t}^{\mathtt{ucb}} - \theta_i)^2}{(\frac{\theta_0}{m} + \theta_i)\theta_0 + \Theta_{S_t}}\mathbb{1}\{i \in S_t, \tau_{i,t} \geq 138x(m+1)^2\theta_{\max}^2\}\mathbb{1}\{\mathcal{E}\}\right].$$

Now, since under the event $\mathcal{E}$ by Lemma 2, $\tau_{i,t} \geq 138x(m+1)^2\theta_{\max}^2$ implies

$$n_{i0,t} \geq 69x(\theta_0 + \theta_i)(m+1)\theta_{\max} \geq 69x(\theta_0 + \theta_i).$$

Therefore, we can apply Lemma 1, which further upper-bounds

$$A_T(i) \leq \sum_{t=1}^{T} \mathbb{E}\left[\left(\frac{2^6(\theta_0 + \theta_i)^2 x}{n_{i0,t}} + \frac{2(22x)^2(\theta_0 + \theta_i)^4}{n_{i0,t}^2(\frac{\theta_0}{m} + \theta_i)}\right)\right.$$
$$\left. \times \frac{\mathbb{1}\{i \in S_t, \tau_{i,t} \geq 138x(m+1)^2\theta_{\max}^2\}}{\theta_0 + \Theta_{S_t}}\mathbb{1}\{\mathcal{E}\}\right]$$

$$\leq \sum_{t=1}^{T} \mathbb{E}\left[\left(\frac{2^6(\theta_0 + \theta_i)^2 x}{n_{i0,t}} + \frac{15x(\theta_0 + \theta_i)^3}{n_{i0,t}\theta_{\max}(\theta_0 + m\theta_i)}\right) \times \frac{\mathbb{1}\{i \in S_t\}}{\theta_0 + \Theta_{S_t}}\mathbb{1}\{\mathcal{E}\}\right]$$

where we used $n_{i0,t} \geq 69x(\theta_0 + \theta_i)m\theta_{\max}$ in the last inequality. Then, we get

$$A_T(i) \leq \sum_{t=1}^{T} \mathbb{E}\left[\left(\frac{(\theta_0 + \theta_i)^2 x}{n_{i0,t}} + \frac{30x(\theta_0 + \theta_i)}{n_{i0,t}}\right) \times \frac{\mathbb{1}\{i \in S_t\}}{\theta_0 + \Theta_{S_t}}\mathbb{1}\{\mathcal{E}\}\right]$$

$$\leq (94 + 64\theta_i)x \sum_{t=1}^{T} \mathbb{E}\left[\frac{(\theta_0 + \theta_i)\mathbb{1}\{i \in S_t\}}{(\theta_0 + \Theta_{S_t})n_{i0,t}}\right]$$

$$= (94 + 64\theta_i)x\mathbb{E}\left[\sum_{t=1}^{T} \frac{\mathbb{1}\{i_t \in \{i,0\}, i \in S_t\}}{n_{i0,t}}\right]$$

$$= (94 + 64\theta_i)x\mathbb{E}\left[1 + \log\left(n_{i0}(T)\right)\right]$$

$$\leq 158\theta_{\max}x(1 + \log T).$$

Substituting into (16), we then obtain using Cauchy-Schwarz inequality,

$$\sum_{t=1}^{T}\mathbb{E}\left[\left(\left(\mathcal{R}(S_t, \theta_t^{\mathtt{ucb}}) - \mathcal{R}(S_t, \theta)\right) \wedge 1\right)\mathbb{1}\{\mathcal{E}\}\right]$$

$$\leq 138xm^2K\theta_{\max}^2 + 13\sqrt{\theta_{\max}x(1 + \log T)}\sum_{i=1}^{K}\sqrt{\sum_{t=1}^{T}\mathbb{E}\left[\frac{(\frac{\theta_0}{m} + \theta_i)\mathbb{1}\{i \in S_t\}}{\theta_0 + \Theta_{S_t}}\right]}$$

$$\leq 138xm^2K\theta_{\max}^2 + 13\sqrt{\theta_{\max}x(1 + \log T)}\sqrt{\mathbb{E}\left[K\sum_{t=1}^{T}\frac{\sum_{i=1}^{K}(\frac{\theta_0}{m} + \theta_i)\mathbb{1}\{i \in S_t\}}{\theta_0 + \Theta_{S_t}}\right]}$$

$$= 138xm^2K\theta_{\max}^2 + 13\sqrt{\theta_{\max}x(1 + \log T)KT}.$$

Finally, replacing into Inequality (15) yields

$$Reg_T^{\mathtt{wtd}} \leq 4T^2Ke^{-x} + 138xm^2K\theta_{\max}^2 + 13\sqrt{\theta_{\max}x(1 + \log T)KT}.$$

Choosing $x = 2\log T$ concludes the proof. $\qquad\square$

### B.6 Proof of Theorem 5

The proof follows the one of Theorem 4, except that the concentration lemmas should be generalized to any pairs $(i,j)$ instead of only with respect to item 0, whose proofs are left to the reader and closely follows the one of Lemma 1 and 2. For simplicity, this proof is performed up to universal multiplicative constants, using the rough inequality $\lesssim$.

**Lemma 9.** *Let $T \geq 1$ and $x > 0$. Then, with probability at least $1 - 3K(K+1)Te^{-x}$, simultaneously for all $t \in [T]$ and $i \neq j$ in $[\tilde{K}]$: $\gamma_{ij} := \frac{\theta_i}{\theta_j} \leq \gamma_{ij,t}^{ucb}$ and one of the following two inequalities is satisfied*

$$n_{ij,t} < 69x(1+\gamma_{ij}) \qquad or \qquad \gamma_{ij,t}^{ucb} \leq \gamma_{ij} + 4(\gamma_{ij}+1)\sqrt{\frac{2\gamma_{ij}x}{n_{ij,t}}} + \frac{22x(\gamma_{ij}+1)^2}{n_{ij,t}}.$$

**Lemma 10.** *Let $T \geq 1$ and $x > 0$. Then, with probability at least $1 - 3K(K+1)Te^{-x}$, simultaneously for all $t \in [T]$ and $i \in [K]$: $\widehat{\theta}_{i,t}^{ucb} := \min_j \gamma_{ij,t}^{ucb}\gamma_{j0,t}^{ucb} \geq \theta_i$ and for all $j$ one of the following two inequalities is satisfied*

$$n_{ij,t} \lesssim x(1+\gamma_{ij}) \qquad or \qquad n_{j0,t} \lesssim x(1+\theta_j)^2\theta_j^{-1}$$

*or*

$$\gamma_{ij,t}^{ucb}\gamma_{j0,t}^{ucb}-\theta_i \lesssim \sqrt{(\gamma_{ij}+1)\theta_i x}\left(\sqrt{\frac{(\theta_i+\theta_j)}{n_{ij,t}}}+\sqrt{\frac{(1+\theta_j)}{n_{j0,t}}}\right)+(\gamma_{ij}+1)\frac{(\theta_i+\theta_j)x}{n_{ij,t}}+\frac{\gamma_{ij}(1+\theta_j)^2x}{n_{j0,t}}.$$

*Proof of Lemma 10.* The proof follows from Lemma 9. If $n_{ij,t} > Cx(1+\gamma_{ij})$ and $n_{j0,t} > Cx(1+\theta_j)$ for some large enough constant C, we have

$$\gamma_{ij,t}^{ucb} \leq \gamma_{ij} + 4(\gamma_{ij}+1)\sqrt{\frac{2\gamma_{ij}x}{n_{ij,t}}} + \frac{22x(\gamma_{ij}+1)^2}{n_{ij,t}}$$

and

$$\gamma_{j0,t}^{ucb} \leq \gamma_{j0} + 4(\gamma_{j0}+1)\sqrt{\frac{2\gamma_{j0}x}{n_{j0,t}}} + \frac{22x(\gamma_{j0}+1)^2}{n_{j0,t}} \leq 2\gamma_{j0}.$$

This implies,

$$\begin{aligned}
\gamma_{ij,t}^{ucb}\gamma_{j0,t}^{ucb} - \theta_i &= \gamma_{ij,t}^{ucb}\gamma_{j0,t}^{ucb} - \gamma_{ij}\gamma_{j0} = (\gamma_{ij,t}^{ucb} - \gamma_{ij})\gamma_{j0,t}^{ucb} + \gamma_{ij}(\gamma_{j0,t}^{ucb} - \gamma_{j0}) \\
&\leq 2(\gamma_{ij,t}^{ucb} - \gamma_{ij})\gamma_{j0} + \gamma_{ij}(\gamma_{j0,t}^{ucb} - \gamma_{j0}) \\
&\leq 8\gamma_{j0}(\gamma_{ij}+1)\sqrt{\frac{2\gamma_{ij}x}{n_{ij,t}}} + \frac{44x\gamma_{j0}(\gamma_{ij}+1)^2}{n_{ij,t}} \\
&\quad + 4\gamma_{ij}(\gamma_{j0}+1)\sqrt{\frac{2\gamma_{j0}x}{n_{j0,t}}} + \frac{22x\gamma_{ij}(\gamma_{j0}+1)^2}{n_{j0,t}}.
\end{aligned}$$

Replacing $\gamma_{ij} = \theta_i/\theta_j$ and $\gamma_{j0} = \theta_j$ concludes the proof. □

**Lemma 11.** *Let $T \geq 1$ and $x > 0$. Then, with probability at least $1 - K(K+1)Te^{-x}$*

$$\tau_{ij,t} < 2x\frac{(\theta_0+\Theta_{S^*})^2}{\theta_i+\theta_j} \quad or \quad n_{ij,t} \geq \frac{(\theta_i+\theta_j)\tau_{ij,t}}{2(\theta_0+\Theta_{S^*})}, \tag{17}$$

*where $\tau_{ij,t} := \sum_{s=1}^{t-1} \mathbb{1}\{\{i,j\} \subseteq S_s\}$ simultaneously for all $t \in [T]$ and $i \neq j \in [K]$.*

*Proof of Theorem 5.* Let $\mathcal{E}$ be the high-probability event of Lemmas 10 and 11 are satisfied, so that $\mathbb{P}(\mathcal{E}) \geq 1 - 4K^2Te^{-x}$. First, note that since we have under the event $\mathcal{E}$, $\widehat{\theta}_t^{ucb} \leq \theta_t^{ucb}$, our procedure also satisfies the regret upper-bound

$$Reg_T^{wtd} \leq O(\sqrt{\theta_{\max}KT\log T})$$

of Theorem 4. Indeed, all upper-bounds of the proof of Theorem 4 remain valid upper-bounds except the probability of the event $\mathcal{E}^c$ which is $O(T^{-1})$ for $x = 2\log T$.

Let us now prove that we also have $R_T \leq O(K\sqrt{T}\log T)$ with no asymptotic dependence on $\theta_{\max}$ when $T \to \infty$.

Then,

$$Reg_T^{\texttt{wtd}} = \sum_{t=1}^T \mathbb{E}\big[\mathcal{R}(S^*,\theta) - \mathcal{R}(S_t,\theta)\big] \tag{18}$$

$$= \sum_{t=1}^T \mathbb{E}\big[(\mathcal{R}(S^*,\theta) - \mathcal{R}(S_t,\theta))\mathbb{1}\{\mathcal{E}\} + (\mathcal{R}(S^*,\theta) - \mathcal{R}(S_t,\theta))\mathbb{1}\{\mathcal{E}^c\}\big]$$

$$\leq \sum_{t=1}^T \mathbb{E}\Big[((\mathcal{R}(S_t,\widehat{\theta}_t^{\texttt{ucb}}) - \mathcal{R}(S_t,\theta)) \wedge \mathcal{R}(S^*,\theta))\mathbb{1}\{\mathcal{E}\} + \mathcal{R}(S^*,\theta)\mathbb{1}\{\mathcal{E}^c\}\Big].$$

Then, using $\mathcal{R}(S^*,\theta) \leq \max_i r_i \leq 1$, we get

$$Reg_T^{\texttt{wtd}} \leq \sum_{t=1}^T \mathbb{E}\Big[((\mathcal{R}(S_t,\widehat{\theta}_t^{\texttt{ucb}}) - \mathcal{R}(S_t,\theta)) \wedge 1)\mathbb{1}\{\mathcal{E}\} + \mathbb{1}\{\mathcal{E}^c\}\Big]$$

$$\leq 4T^2 K(K+1)^2 e^{-x} + \sum_{t=1}^T \mathbb{E}\Big[\big((\mathcal{R}(S_t,\widehat{\theta}_t^{\texttt{ucb}}) - \mathcal{R}(S_t,\theta)) \wedge 1\big)\mathbb{1}\{\mathcal{E}\}\Big]. \tag{19}$$

Follow the proof of Theorem 4, we upper-bound the second term of the right-hand-side of (19):

$$\sum_{t=1}^T \mathbb{E}\Big[\big((\mathcal{R}(S_t,\widehat{\theta}_t^{\texttt{ucb}}) - \mathcal{R}(S_t,\theta)) \wedge 1\big)\mathbb{1}\{\mathcal{E}\}\Big] \tag{20}$$

$$= \sum_{t=1}^T \mathbb{E}\bigg[\bigg(\bigg(\min_{j\in[K]} \sum_{i\in S_t} \frac{r_i\widehat{\theta}_{i,t}^{\texttt{ucb}}}{1 + \sum_{j\in S_t}\widehat{\theta}_{j,t}^{\texttt{ucb}}} - \frac{r_i\theta_i}{1 + \sum_{j\in S_t}\theta_j}\bigg) \wedge 1\bigg)\mathbb{1}\{\mathcal{E}\}\bigg]$$

$$\leq \sum_{t=1}^T \mathbb{E}\bigg[\bigg(\bigg(\sum_{i\in S_t} \frac{r_i(\widehat{\theta}_{i,t}^{\texttt{ucb}} - \theta_i)}{\theta_0 + \Theta_{S_t}}\bigg) \wedge 1\bigg)\mathbb{1}\{\mathcal{E}\}\bigg] \qquad \text{because } \sum_{i\in S_t}\widehat{\theta}_{i,t}^{\texttt{ucb}} \geq \Theta_{S_t} \text{ under } \mathcal{E}$$

$$\leq \sum_{t=1}^T \mathbb{E}\bigg[\bigg(\bigg(\sum_{i\in S_t} \frac{|\widehat{\theta}_{i,t}^{\texttt{ucb}} - \theta_i|}{\theta_0 + \Theta_{S_t}}\bigg) \wedge 1\bigg)\mathbb{1}\{\mathcal{E}\}\bigg] \qquad \text{because } r_i \leq 1$$

$$\leq \sum_{i=1}^K \mathbb{E}\bigg[\sum_{t=1}^T \bigg(\frac{|\widehat{\theta}_{i,t}^{\texttt{ucb}} - \theta_i|}{\theta_0 + \Theta_{S_t}} \wedge 1\bigg)\mathbb{1}\{i\in S_t\}\mathbb{1}\{\mathcal{E}\}\bigg]$$

$$\leq \sum_{i=1}^K \mathbb{E}\bigg[\sum_{t=1}^T \bigg(\frac{|\gamma_{ij_t,t}^{\texttt{ucb}}\gamma_{j_t 0,t}^{\texttt{ucb}} - \theta_i|}{\theta_0 + \Theta_{S_t}} \wedge 1\bigg)\mathbb{1}\{i\in S_t\}\mathbb{1}\{\mathcal{E}\}\bigg]$$

where $j_t = \text{argmax}_{j\in S_t\cup\{0\}} \theta_j$, where the last inequality is by definition of $\widehat{\theta}_{i,t}^{\texttt{ucb}}$. Now, from Lemma 10, paying an additive exploration cost to ensure that $n_{ij,t} \gtrsim x(1 + \gamma_{ij})$ and $n_{j0,t} \gtrsim x(1 + \theta_j)^2\theta_j$ for all $j \in S_t$ such that $\theta_j \geq \theta_0$. From Lemma 11, this is satisfied if for some constant $C > 0$

$$\tau_{ij,t} > Cm^2\theta_{\max}^2 x.$$

Such a condidtion can be wrong for a couple $(i,j) \in S_t^2$ at most during $CK^2 m^2 \theta_{\max}^2 x = O(\log T)$ rounds (since $\tau_{ij,t}$ increases then). Thus, for $C$ large enough,

$$\sum_{t=1}^T \mathbb{E}\Big[\big((\mathcal{R}(S_t,\widehat{\theta}_t^{\texttt{ucb}}) - \mathcal{R}(S_t,\theta)) \wedge 1\big)\mathbb{1}\{\mathcal{E}\}\Big]$$

$$\leq O(\log T) + \sum_{i=1}^{K} \mathbb{E}\left[\sum_{t=1}^{T} \frac{|\gamma_{ij_t,t}^{\mathbf{ucb}}\gamma_{j_t0,t}^{\mathbf{ucb}} - \theta_i|}{\theta_0 + \Theta_{S_t}} \mathbb{1}\{i \in S_t, \tau_{ij_t,t} \wedge \tau_{j_t,t} \geq Cxm^2\theta_{\max}^2\}\mathbb{1}\{\mathcal{E}\}\right]$$

$$\lesssim O(\log T) + \sum_{i=1}^{K} \mathbb{E}\left[\sum_{t=1}^{T} \left(\sqrt{(\gamma_{ij_t}+1)\theta_i x}\left(\sqrt{\frac{(\theta_i+\theta_{j_t})}{n_{ij_t,t}}} + \sqrt{\frac{(1+\theta_j)}{n_{j_t0,t}}}\right)\right.\right.$$
$$\left.\left.+ (\gamma_{ij_t}+1)\frac{(\theta_i+\theta_{j_t})x}{n_{ij_t,t}} + \frac{\gamma_{ij_t}(1+\theta_{j_t})^2 x}{n_{j_t0,t}}\right)\frac{\mathbb{1}\{i\in S_t\}}{\theta_0+\Theta_{S_t}}\right]$$

$$\leq O(\log T) + \sum_{i=1}^{K}\mathbb{E}\left[\sum_{t=1}^{T}\sqrt{(\gamma_{ij_t}+1)\theta_i x}\left(\sqrt{\frac{(\theta_i+\theta_{j_t})}{n_{ij_t,t}}} + \sqrt{\frac{(1+\theta_{j_t})}{n_{j_t0,t}}}\right)\frac{\mathbb{1}\{i\in S_t\}}{\theta_0+\Theta_{S_t}}\right]$$

where the last inequality is because using that $\{i, j_t, 0\} \subseteq S_t$, we have

$$\mathbb{E}\left[\sum_{t=1}^{T}\frac{1+\theta_{j_t}}{(1+\Theta_{S_t})n_{j_t0,t}}\right] = \mathbb{E}\left[\sum_{t=1}^{T}\sum_{j=1}^{K}\frac{\mathbb{1}\{i_t\in\{j,0\}\}}{n_{j0,t}}\mathbb{1}\{j=j_t\}\right] \leq K(1+\log T).$$

and

$$\mathbb{E}\left[\sum_{t=1}^{T}\frac{\theta_i+\theta_{j_t}}{(1+\Theta_{S_t})n_{ij_t,t}}\right] = \mathbb{E}\left[\sum_{t=1}^{T}\sum_{j=1}^{K}\frac{\mathbb{1}\{i_t\in\{j,i\}\}}{n_{j0,t}}\mathbb{1}\{j=j_t\}\right] \leq K(1+\log T).$$

Then, by Cauchy-Schwarz inequality we further get

$$\sum_{t=1}^{T}\mathbb{E}\left[\left((\mathcal{R}(S_t,\widehat{\theta}_t^{\mathbf{ucb}}) - \mathcal{R}(S_t,\theta))\wedge 1\right)\mathbb{1}\{\mathcal{E}\}\right]$$

$$\lesssim O(\log T) + \sum_{i=1}^{K}\sqrt{\mathbb{E}\left[\sum_{t=1}^{T}\frac{(\gamma_{ij_t}+1)\theta_i\mathbb{1}\{i\in S_t\}x}{\theta_0+\Theta_{S_t}}\right]} \tag{21}$$
$$\times\sqrt{\mathbb{E}\left[\sum_{t=1}^{T}\left(\frac{(\theta_i+\theta_{j_t})}{n_{ij_t,t}} + \frac{(1+\theta_{j_t})}{n_{j_t0,t}}\right)\frac{\mathbb{1}\{i\in S_t\}}{\theta_0+\Theta_{S_t}}\right]}$$

$$\lesssim O(\log T) + \sum_{i=1}^{K}\sqrt{\mathbb{E}\left[\sum_{t=1}^{T}\frac{(\gamma_{ij_t}+1)\theta_i\mathbb{1}\{i\in S_t\}x}{\theta_0+\Theta_{S_t}}\right]}\sqrt{K\log T}$$

$$\lesssim O(\log T) + \sum_{i=1}^{K}\sqrt{\mathbb{E}\left[\sum_{t=1}^{T}\frac{\theta_i\mathbb{1}\{i\in S_t\}x}{\theta_0+\Theta_{S_t}}\right]}\sqrt{K\log T} \text{ (because } \gamma_{ij_t}\leq 1 \text{ by definition of } j_t)$$

$$\leq O(K\sqrt{Tx\log T}) = O(K\sqrt{T}\log T), \tag{22}$$

where the last inequality is by Jensen's inequality and the equality by setting $x = 2\log T$ to control the probability that $\mathcal{E}^c$ occurs. This concludes the proof. $\qquad\square$

