^{\mathrm{ucb}}_{i,t} - \theta_i|}{\theta_0 + \Theta_{S_t}}\right)\mathbb{1}\{i \in S_t, \mathcal{E}\}\right]$$

$$\lesssim O(\tau_0) + \sum_{i=1}^{K} \mathbb{E}\left[\sum_{t=1}^{T} \frac{|\theta^{\mathrm{ucb}}_{i,t} - \theta_i|}{\theta_0 + \Theta_{S_t}}\mathbb{1}\{i \in S_t, \tau_{i,t} \geq \tau_0, \mathcal{E}\}\right]$$

$$\lesssim O(\tau_0) + \sum_{i=1}^{K} \sqrt{\sum_{t=1}^{T} \mathbb{E}\left[\frac{\theta_i \mathbb{1}\{i \in S_t\}}{\theta_0 + \Theta_{S_t}}\right]} \times \underbrace{\sqrt{\sum_{t=1}^{T} \mathbb{E}\left[\left(\frac{\theta^{\mathrm{ucb}}_{i,t} - \theta_i}{\theta_0 + \Theta_{S_t}}\right)^2 \frac{\theta_0 + \Theta_{S_t}}{\theta_i}\mathbb{1}\{i \in S_t, \tau_{i,t} \geq \tau_0, \mathcal{E}\}\right]}}_{=:A_T(i)} \tag{6}$$

where the last inequality is by Cauchy-Schwarz inequality. Now, the term $A_T(i)$ above may be upper-bounded using Lemmas 1 and 2,

$$A_T(i) = \mathbb{E}\left[\frac{(\theta^{\mathrm{ucb}}_{i,t} - \theta_i)^2}{\theta_i(\theta_0 + \Theta_{S_t})}\mathbb{1}\{i \in S_t, \tau_{i,t} \geq \tau_0, \mathcal{E}\}\right] \lesssim \sum_{t=1}^{T} \mathbb{E}\left[\frac{(\theta_0 + \theta_i)^2 x}{n_{i0,t}(\theta_0 + \Theta_{S_t})}\mathbb{1}\{i \in S_t\}\right]$$

$$\lesssim \theta_{\max} x \sum_{t=1}^{T} \mathbb{E}\left[\frac{(\theta_0 + \theta_i)\mathbb{1}\{i \in S_t\}}{(\theta_0 + \Theta_{S_t})n_{i0,t}}\right] = \theta_{\max} x \mathbb{E}\left[\sum_{t=1}^{T} \frac{\mathbb{1}\{i_t \in \{i,0\}, i \in S_t\}}{n_{i0,t}}\right] \lesssim \theta_{\max} x \log T$$

where in the last inequality we used that $\sum_{n=1}^{T} n^{-1} \leq 1 + \log T$. Substituting into (6), Jensen's inequality entails,

$$\sum_{t=1}^{T} \mathbb{E}\Big[\big(\mathcal{R}(S_t, \theta^{\mathrm{ucb}}_t) - \mathcal{R}(S_t, \theta)\big)\mathbb{1}\{\mathcal{E}\}\Big] \lesssim O(\tau_0) + \mathbb{E}\left[\sqrt{\theta_{\max} x \log T} \sum_{i=1}^{K} \sqrt{\sum_{t=1}^{T} \frac{\theta_i \mathbb{1}\{i \in S_t\}}{\theta_0 + \Theta_{S_t}}}\right]. \tag{7}$$

The proof is finally concluded by applying Cauchy-Schwarz inequality which yields:

$$\sum_{i=1}^{K} \sqrt{\sum_{t=1}^{T} \frac{\theta_i \mathbb{1}\{i \in S_t\}}{\theta_0 + \Theta_{S_t}}} \leq \sqrt{K \sum_{t=1}^{T} \frac{\sum_{i=1}^{K} \theta_i \mathbb{1}\{i \in S_t\}}{\theta_0 + \Theta_{S_t}}} \leq \sqrt{KT}.$$

Finally, combining the above result with (5) and (7) concludes the proof

$$Reg^{\mathrm{wtd}}_T \lesssim TP(\mathcal{E}^c) + O(\tau_0) + \sqrt{\theta_{\max} x K T \log T}.$$

Choosing $x = 2\log T$ ensures $TP(\mathcal{E}^c) \leq O(1)$ and $\tau_0 \leq O(\log T)$. $\qquad\square$

## 4 Improved dependance on $\theta_{\max}$ with Adaptive Pivot Selection

A problem with Algorithm 1 stems from estimating all $\theta_i$ based on pairwise comparisons with item 0. When $\theta_{\max} \gg \theta_0 = 1$, item 0 may not be sampled enough as the winner, leading to poor estimators. This deficiency contributes to the suboptimal dependence on $\theta_{\max}$ observed in Theorems 3 and 4 and in prior work, such as Agrawal et al. (2019). We propose the following fix to optimize the pivot. For all $i, j \in [K] \cup \{0\}$ we define $\gamma_{ij} = \frac{\theta_i}{\theta_j}$,

$$\gamma^{\mathrm{ucb}}_{ij,t} = p^{\mathrm{ucb}}_{ij,t}/(1 - p^{\mathrm{ucb}}_{ij,t})_+ \qquad \text{and} \qquad \gamma^{\mathrm{ucb}}_{ii,t} = 1,$$

where $p^{\mathrm{ucb}}_{ij,t}$ are defined in (3). For all rounds $t$, the algorithm AOA-RB$_{\mathrm{PL}}$-Adaptive selects

$$S_t = \underset{|S| \leq m}{\operatorname{argmax}} \mathcal{R}(S, \widehat{\theta}^{\mathrm{ucb}}_t) \qquad \text{where} \qquad \widehat{\theta}^{\mathrm{ucb}}_{i,t} := \min_{j \in [K] \cup \{0\}} \gamma^{\mathrm{ucb}}_{ij,t}\gamma^{\mathrm{ucb}}_{j0,t}.$$

With the above definition of $\widehat{\theta}_{i,t}^{\text{ucb}}$, any item $i$ is compared to the base item 0 through the best possible item $j$. When $j$ is a strong item that is often selected both $\gamma_{ij,t}^{\text{ucb}}$ and $\gamma_{j0,t}^{\text{ucb}}$ are sharp upper-bounds of $\gamma_{ij}$ and $\gamma_{j0}$, making $\widehat{\theta}_{i,t}^{\text{ucb}}$ itself a sharp upper-confidence bound for $\theta_i$. This definition in turn also satisfies the condition $\widehat{\theta}_{i,t}^{\text{ucb}} \geq \theta_i$ required by Lemma 8 and crucial for our analysis. The condition would not hold if we used $\gamma_{ij,t}^{\text{ucb}}$ directly without multiplying it with $\gamma_{j0,t}^{\text{ucb}}$.

We offer below a regret bound that underscores the value of optimizing the pivot when $\theta_{\max} \gg K$. Note that while the algorithm and analysis are presented for the weighted objective with winner feedback only, it can be adapted to other objectives by replacing $\mathcal{R}(S, \theta)$ with the new objective in the analysis, as long as Lemma 8 remains valid.

**Theorem 5.** *Let $\theta_{\max} \geq 1$. For any $\theta \in [0, \theta_{\max}]^K$ and weights $\mathbf{r} \in [0, 1]^K$, the weighted regret of AOA-RB$_{PL}$-Adaptive is upper-bounded as*

$$Reg_T^{wtd} = O\big(\sqrt{\min\{\theta_{\max}, K\} K T} \log T\big)$$

*as $T \to \infty$ for the choice $x = 2 \log T$ (when defining $p_{ij,t}^{ucb}$).*

**Remark 1** (Drastic Improvement over Prior Works). *Asymptotically, when $\theta_{\max}$ is constant, the regret is $O(K\sqrt{T}\log T)$, eliminating any dependence on $\theta_{\max}$. This allows for handling scenarios where the No-Choice item is highly unlikely, which is not achievable in previous works such as Agrawal et al. (2019; 2017). Agrawal et al. (2019) did attempt in their Thm. 4 to relax the assumption of $\theta_{\max} = \theta_0$ and shows a bound of order $O\big(\max\{\theta_{\max}/\theta_0, 1\}^{1/2}\sqrt{KT}\big)$, which unfortunately blows to $\infty$ as $\theta_0 \to 0$ or equivalently $\theta_{\max} \to \infty$, leading to a vacuous bound. Here, lies the stark improvement and one of the key contributions, as also corroborated in our experimental evaluation Sec. 5 (Fig. 2).*

**Remark 2** (Beyond MNL Assortment: Extending to any general RUM based Choice Models). *Although, in this paper, we primarily focused on MNL based choice models, it is worth mentioning that our proposed algorithms can be generalized to more general random utility-based models (RUMs) Azari et al. (2012b); Saha & Ghoshal (2022) pursuing the ideas from Saha & Gopalan (2020) that extends the RB based parameter estimation technique to any RUM($\boldsymbol{\theta}$) choice models: Precisely, using the RB based RUM-parameter estimation technique of Saha & Gopalan (2020), we can show a regret bound of $\tilde{O}(\frac{\sqrt{\min(\theta_{\max}, K)}}{c_{rum}}\sqrt{KT})$ for our proposed algorithm AOA-RBPL, where $c_{rum}$ is the parameter associated to the minimum advantage ratio (min-AR) of the underlying RUM($\boldsymbol{\theta}$) model, as defined in Thm6 of Saha & Gopalan (2020). In particular, $c_{rum}$ can shown to be a constant given a fixed RUM model, e.g. $c_{rum} = 1/4$ for Exp(1), Gamma(2, 1), $c_{rum} = 1/(4\sigma)$ for Gumbel($\mu, \sigma$), $c_{rum} = \lambda/4$ for Weibull($\lambda, 1$), $c_{rum} = 1/3$ for Gaussian(0, 1), etc (using Cor5 of Saha & Gopalan (2020)).*

Our algorithms and analyses thus apply to any general RUM($\boldsymbol{\theta}$) based choice models; we stick to the special case of MNL models in this paper for brevity and keep the main focus on the AOA problem and the related algorithmic novelties.

The proof of Theorem 5 is deferred to the App. B, with a key step relying on selecting the pivot $j_t = \arg\max_{j \in S_t \cup \{0\}} \theta_j$. The use of $|\widehat{\theta}_{i,t}^{\text{ucb}} - \theta_i| \leq |\gamma_{ij_t,t}^{\text{ucb}} - \theta_i|$ provides confidence upper-bounds with an improved dependence on $\theta_{\max}$, leveraging the fact that $\theta_{j_t} \geq \theta_i$. Due to the varying pivot over time, a telescoping argument introduces an additive factor $\sqrt{K}$.

## 5 EXPERIMENTS

We run experiments to compare the performance of our method with the state-of-the-art methods. All results are averaged across 100 runs. We evaluate the performance of our main algorithm AOA-RB$_{PL}$-Adaptive (Sec. 4), referred as "Our Alg-1 (Adaptive Pivot)", with the following algorithms: AOA-RB$_{PL}$ (Sec. 3) referred as "Our Alg-2 (No-Choice Pivot)", and MNL-UCB, the state-of-the-art algorithm for AOA (Agrawal et al. (2019), Alg. 1).

**Different PL ($\boldsymbol{\theta}$) Environments.** We report our experiment results on two datasets with $K = 50$ items: (1) Arith50 with PL parameters $\theta_i = 1 - (i - 1)0.02$, $\forall i \in [50]$. (2) Bad50 with PL parameters $\theta_i = 0.6$, $\forall i \in [50] \setminus \{25\}$ and $\theta_{25} = 0.8$. For simplicity of computing the assortment choices $S_t$, we assume $r_i = 1$, $\forall i \in [K]$.

**(1). Averaged Regret with weak NC** ($\theta_{\max}/\theta_0 \gg 1$) **(Fig. 1):** In our first experiment, we set $m = 5$ and $\theta_0/\theta_{\max} = 0.01$ and report the average regret of the above three algorithms for our two objectives.

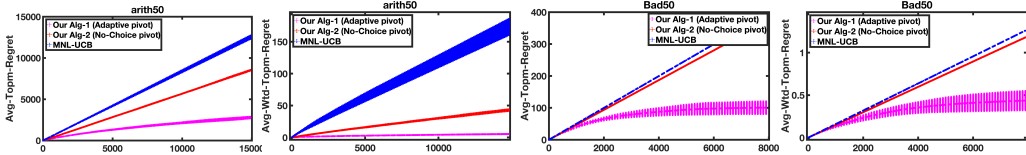

Figure 1: Averaged Regret for $m = 5$, $\theta_0 = 0.01$

Fig. 1 shows that our algorithm AOA-RB$_{\text{PL}}$-Adaptive (with adaptive pivot) significantly outperforms the other two algorithms, while our algorithm AOA-RB$_{\text{PL}}$ with no-choice (NC) pivot still outperforms MNL-UCB.

**(2). Averaged Regret vs No-Choice PL Parameter** ($\theta_{\max}/\theta_0$) **(Fig. 2):** In this experiment, we evaluate the regret performance of our algorithm AOA-RB$_{\text{PL}}$-Adaptive. We report the experiment on Artith50 PL dataset and set the subsetsize $m = 5$, $\theta_{\max}/\theta_0 = \{1, 0.5, 0.1, 0.05, 0.01, 0.005, 0.001\}$. Fig. 2 shows the increase in the performance gap between our algorithm AOA-RB$_{\text{PL}}$-Adaptive (with adaptive pivot) with decreasing $\theta_0/\theta_{\max}$.

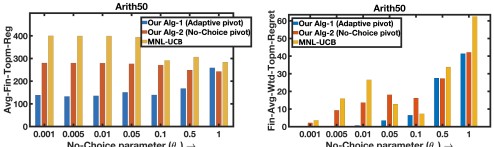

Figure 2: Comparative performance for varying $\theta_0/\theta_{\max}$, $m = 5$

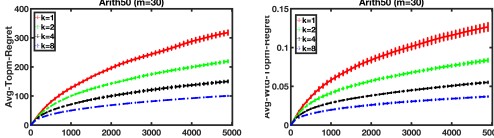

Figure 3: Tradofff: Averaged Regret vs length of the $k$ rank-ordered feedback

**(3). Averaged Regret vs Length of the rank-ordered feedback ($k$) (Fig. 3):** We also run a thought experiment to understand the tradeoff between learning rate with $k$-length rank-ordered feedback, where given any assortment $S_t \subseteq [K]$ of size $m$, the learner gets to see the top-$k$ draws ($k \leq m$) from the PL model without replacement. This is a stronger feedback than the winner (i.e. top-1 for $k = 1$) feedback and, as expected, we see in Fig. 3 an improved regret (for both notions) when increasing $k$. The experiment are run on the Artith50 dataset with $m = 30$ and $k \in \{1, 2, 4, 8\}$.

## 6 CONCLUSION

We address the Active Optimal Assortment Selection problem with PL choice models, introducing a versatile framework (*AOA*) that eliminates the need for a strong default item, typically assumed as the No-Choice (NC) item in the existing literature. Our proposed algorithms employ a novel 'Rank-Breaking' technique to establish tight concentration guarantees for estimating the parameters of the PL model. Our approach stands out for its practicality and avoids the suboptimal practice of repeatedly selecting the same set of items until the default item prevails. This is beneficial when the default item's quality ($\theta_0$) is significantly lower than the quality of the best item ($\theta_{\max}$). Our algorithms are computationally efficient, optimal (up to log factors), and free from restrictive assumptions on the default item.

**Future Works.** Among many interesting questions to address in the future, it will be interesting to understand the role of the No-Choice (NC) item in the algorithm design, precisely, can we design efficient algorithms without the existence of NC items with a regret rate still linear in $\theta_{\max}$? Further, it will be interesting to extend our results to more general choice models beyond the PL model Chen et al. (2021); Désir et al. (2016a;b). What is the tradeoff between the subsetsize $m$ and the regret for such general choice models? Extending our results to large (potentially infinite) decision spaces and contextual settings would also be a very useful and practical contribution to the literature of assortment optimization.