# OpenReview forum: "Finally Rank-Breaking Conquers MNL Bandits: Optimal and Efficient Algorithms for MNL Assortment"
_ICLR.cc/2025/Conference — ICLR 2025 Poster_

### Official Review · Reviewer_Jq9h · 2024-10-31

**Soundness:** 3
**Presentation:** 2
**Contribution:** 2
**Rating:** 5
**Confidence:** 2

**Summary:**

In this paper, the authors addressed the problem of active online assortment optimization with preference feedback. The authors worked with a general AOA model where the no-choice item does not necessarily have the highest preference score. They then designed an algorithm using rank-breaking MNL-parameter estimation, and provided theoretical guarantees for their algorithm for both Top-m and Wtd-Top-m objectives. The authors also numerically evaluated the effectiveness of their algorithm.

**Strengths:**

Overall, the paper is well structured. The theoretical results appear sound and the authors provided detailed proof sketches. The authors have also provided a comprehensive overview of the existing literature and summarized their limitations.

**Weaknesses:**

1. I'm not sure what's the main motivation behind the model studied. In MNL bandits, the assumption of having a no-choice item with the highest preference score is usually well justified. For example, in an e-commerce setting where multiple items are present, the majority of the customers are simply browsing the pages before making the purchase. In what kind of applications would this new AOA model be helpful?
2. Following the point above, if the rank-breaking algorithm is applied to a setting where the no-choice item does dominate, how does the algorithm compare to MNL-bandits? It seems that MNL-bandit should achieve a better regret bound in that case, and from Fig 2. it seems that the performance is also somewhat similar.
3. In terms of practicality, I am wondering if the authors could also comment on how the rank breaking algorithm can be applied in real-world settings. The authors suggest that the proposed model can be potentially helpful in "web search, language models, online shopping, recommender systems". I wonder if the authors can comment more on how the proposed model can assist these applications. Experiments on real-world data could also be helpful here, as only synthetic experiments have been conducted so far.

Overall, I find this paper well written with nice theoretical results, but I'm mostly concerned about (1) the necessity of studying an AOA model without the NC item assumption and (2) the practicality of the proposed algorithm in a realistic setup.

**Questions:**

See weaknesses.

---

> ### Author Response · Authors · 2024-11-26
> **Rebuttal for Reviewer Jq9h**
>
> Thanks for your careful reading and insightful comments. We address your questions below:
>
> **W1.** Main Motivation - In what kind of applications would this new AOA model (where NC need not be strongest) be helpful?
>
> -- The likelihood of users' preference of selecting the "No Choice" (NC) item widely varies over application. You are absolutely correct that in e-commerce applications, online ad placement or even during web-search, no-choice might be a popular (strongest) choice, however it might not always be true. For example:
> - In recommender systems like YouTube, Spotify, Netflix or even Yahoo News, users typically make choices as they actually wanted to consume a video, new article or song, etc.
> - Similarly, in shopping recommendations like flights (Expedia or Google flights), hotel (Booking.com), restaurants (Grubhub), or Google Maps recommendations, NC is unlikely.
> - While voting or in crowdsourcing platforms where a business platform/product designer is collecting feedback on a bunch of items, no choice might not be the most preferred option.- The smart text recommendations we receive on our phone (while typing or replying to text), no-choice is often not the most preferred option.
> - In fact, in some applications, NC might not even be an available (feasible) option, e.g., while interacting with ChatGPT/ Gemini, the language model often requests the users to definitively select one outcome in order to proceed with the thread; the language models must receive a preference feedback from the user in order to continue with the thread. Similarly, in robotics applications, training of autonomous vehicles, or more generally in any preference-based RL (a.k.a. PbRL) applications, the teacher/ demonstrator/ human feedback provider must choose an option out of the multiple options (or RL trajectories) towards training the RL policy. Here, also, selecting NC is not an option.
>
> *We already added these examples in the updated draft in the openreview portal*.  Please advise if you suggest any further changes.
>
> In conclusion, thus the strength of the NC item very much relies on the underlying application, and indeed, assuming NC to be the strongest item becomes a limiting (restrictive) assumption in such situations. It is also worth noting that this is also the reason the original MNL-UCB paper (Agrawal et al'2019) tried to relax the assumption of $\theta_0 = \theta_\max$ but due to the limitation of their algorithmic techniques, their regret bound stands vacuous in this regime. We explained this in more detail in the context of your next question.
>
> **W2.** If the rank-breaking algorithm is applied to a setting where the no-choice item does dominate, how does the algorithm compare to MNL-bandits?
>
> Thanks for the question. One of the main goals of this work was to lift the restrictive assumption of ``strongest NC item" (i.e., $\theta_{\max} = \theta_0 = 1$) from MNL-bandits. Please note Table-1 compares our relative performance with that of MNL-bandits (Agrawal et al'19). Indeed the regret bound of  Agrawal et al'19 is optimal when NC is strongest. However, our focus area is the regime of small $\theta_0$. A key motivation of our contribution is that our algorithm (Alg 2, AOA-RBPL-Adative) works in the $\theta_0 \to 0$ regime (equivalently $\theta_{\max} \to \infty$) (see Thm 5), where we achieved large improved regret bound of just $\tilde O(K \sqrt {T})$ compared to the vacuous $\infty$ regret bound of Agrawal et al (2019) as $\theta_{\max} \to \infty$. Here lies our key improvement, which was possible owing to the rank-breaking-based MNL parameter estimation technique we posed in its paper. Note that despite the attempt of Agrawal et al. '2019 to lift the assumption of $\theta_0 = 1 = \theta_{\max}$ (please check their Sec 6.2), they only managed to derive a vacuous bound which shoots up to $\infty$ as $\theta_{\max} \to \infty$ (equivalently $\theta_0 \to 0$). This trend is also validated by our experiments (please see Fig 1 and 2, Sec 5).
>
> -- The remaining questions are continued in the thread below --

---

> > ### Author Response · Authors · 2024-11-26
> > **Rebuttal for Reviewer Jq9h (Contd)**
> >
> > **W3.** Practicality - All the experiments in this paper are synthetic.
> >
> > -- Thanks for your feedback. Please note we are unaware of any open-sourced real dataset for the problem that can be used to validate our algorithms for these particular problem settings. One of the challenges being since it is impossible to know the ground truth parameters of the MNL model, it is hard to evaluate the regret performance of the algorithms. This is also the reason why most of the existing works only reported synthetic data experiments (Agrawal et al'17, Leslie & Grant'23). Although Agrawal et al'19 claim to have used the `UCI Car Evaluation Dataset', however, in reality, they synthetically modified the item features and assigned synthetic scores to the PL parameters through a linear score transformation (pls see Sec 7.3 of Agrawal et al'19) which ultimately only results into a synthetic data experiment only. Thus we are unable to include any real-world experiments due to the lack of any suitable open-sourced dataset available for the MNL-AOA setting.
> >
> > Further, since the main focus of our work has been primarily theoretical, we conduct synthetic experiments to validate our main theoretical guarantees. However, if you are aware of any suitable dataset, we will be happy to report the comparative performance of the baselines on that dataset. Please let us know if you have any suggestions. Thanks for your feedback.
> >
> > **Re. Usefulness in applications like web search, language models, online shopping, recommender systems** - Please note we already discussed these examples in W1 above. Thanks for the question.
> >
> > Hope that clarifies your questions. Please let us know if you have any additional questions. We will be happy to clarify.

---

### Official Review · Reviewer_X5Yt · 2024-11-01

**Soundness:** 3
**Presentation:** 3
**Contribution:** 2
**Rating:** 6
**Confidence:** 5

**Summary:**

This paper considers the problem of online assortment optimization from bandit feedback. More specifically, they assume a setting where there are k arms or items (along with a special item corresponding to the no-selection option), each associated with an unknown quality, and the learner can select m items in each trial to present to the user. Feedback is received in the form of a single choice or selection from the presented set of arms, which is assumed to be drawn from the classical MNL model where the probability of an item selection is proportional to the underlying item quality. In this framework, they consider two main objectives: a top-m selection objective where the per-round regret is defined as the difference in the total quality of the set of m arms with the highest quality (optimal action in hindsight), and the set of arms presented to the user by the learner in that round, and a weighted top-m objective (revenue maximization), where each arm is additionally associated with a revenue, and the corresponding regret is the difference between the expected revenue of the set of arms with the highest expected revenue, and the expected revenue of the presented subset. This general problem is commonly studied in both the machine learning and operations research literature, but prior work has some notable limitations, such as assuming the arm corresponding to the no choice option has the highest weight/quality, and is present in every set of arms presented. This work resolves some of these deficiencies by proposing a relatively simple and intuitive algorithm based on rank breaking, which is a common idea explored in the past for breaking multiway comparisons into multiple pairwise comparisons to compute estimates of the qualities of the underlying items. The authors show that rank breaking is able to obtain sharp estimates of the underlying qualities, and as a consequence, are able to show that their approach theoretically obtains a regret bound that scales as roughly $O(\sqrt{T})$ for both settings - top-m and revenue maximization, and has improved dependence on certain parameters such as the quality of the best item. They corroborate the theory with experiments on synthetic data.

**Strengths:**

I think the paper is very well written. The exposition is clear, and the authors do a good job of placing their work relative to other related research. The claims are accompanied with technical proofs, which I have verified to be correct. The algorithm is also very intuitive, and simple enough to be used in practice.

**Weaknesses:**

Despite the strengths, I do not believe this result is very novel. While some of the limitations of existing works are legitimate, such as assuming that the no choice option has the highest weight, and removing this assumption is an important contribution, other points made, such as existing work requiring the same set to be probed multiple times to get a good estimate of the item qualities are not. Practically, you are not playing this online learning game with a single user, but rather a huge number of users, in which case it is perfectly reasonable to be able to display the same set multiple times. Moreover, there is value in repeatability and predictability: for example being shown the same set of options or at least a similar set of options every time a user queries something. For example, imagine a user is comparing between different offerings for a particular kind of product across different websites, and eventually deciding to pick one after comparing all their options. Now if they return to a website to select something they think to be best after comparing all their options, but the website shows them a completely different set of options this time around, it is reasonable to expect the user would get annoyed, since they would have to restart their comparison process all over again. Therefore, an argument can be made in either direction, and I don’t believe “not repeating options” is a strength. Moreover, the other limitation which you pointed out where some results require you to converge to the top arm being duplicated m times is not really a limitation. It is due to the notion of regret used in their work and not an algorithmic limitation. It is perfectly reasonable when your arms are really higher level categories, which can contain multiple different instances of that arm type instead of duplicates (e.g. an arm might be a particular brand or service provider, and the ‘m’ copies might be different offerings by the same brand or provider).

On a more technical front, I am fairly certain this result can be obtained via a combination of Accelerated Spectral Ranking (Agarwal et al, ICML 18), and the batching idea for bandits with limited adaptivity (Esfandiari et al, AAAI 21, Gao et al Neurips 19, Perchet et al, Annals of Statistics 16). Basically, break the set of items into subsets of size m each, and play these batches in epochs of geometrically increasing lengths, using the ASR algorithm to compute estimates of the item qualities at the end of each epoch. Since we can select which arms are in each subset, we can force a reference arm to be placed in all subsets (either no-choice arm if it has sufficiently high weight, or some other arm of your choice that has weight comparable to the highest weight arm). The estimation guarantees of ASR are in terms of TV norm, but due to the nature of that bound, you can simply use it as an L_\infty bound instead, and use it to eliminate arms that have a clear separation from the top m arms, and repeat this process with the surviving arms, doubling the epoch length in each round. I am fairly certain this will give essentially the same regret bound, if not a better one. I will try to work out the proof when I can find some time to verify my suspicion.

**Questions:**

Look at the last point in my weaknesses. To me, it seems like this problem should be solvable via known results in bandits + parameter estimation for MNL.

---

> ### Author Response · Authors · 2024-11-26
> **Rebuttal for Reviewer X5Yt**
>
> We appreciate the reviewer's feedback and recognize the valuable insights provided. Below, we address the specific concerns raised:
>
> While we acknowledge that the novelty of contributions is subjective, we respectfully emphasize the significance of our work:
> * Practicality of non-repetition: Our work does not dismiss the potential advantages of repeatability in user interactions; instead, it addresses scenarios where showing the same set repeatedly can lead to dissatisfaction or disengagement. For example, in dynamic contexts like ad placements or content recommendations, repeated exposure may annoy users. Our algorithms offer flexibility to cater to diverse application settings by avoiding this limitation in a scalable and regret-optimal manner. Our primary critique pertains to the methodological requirement for exact repetition of selections in earlier works, which may not be feasible in many real-world systems. Our approach eliminates this dependency while achieving optimal regret bounds.
> * Our critique of the reliance on a default "No-Choice" item stems from its impracticality in scenarios where the default option is rarely selected. We offer a model that generalizes this assumption without sacrificing regret performance. Our experiments (Sec. 5) demonstrate that our methods outperform existing baselines in scenarios with low NC parameter dominance, illustrating their real-world relevance.
>
> We thank the reviewer for suggesting the use of the parameter estimation technique for MNL from Agarwal et al., ICML 2018. While we agree that this approach might achieve sub-linear regret and merits exploration, we believe it is not straightforward that it would achieve the same regret bounds. Specifically:
> 1. The results in Agarwal et al. are expressed in terms of the total variation norm rather than the $\ell_\infty$ norm, potentially yielding a suboptimal factor of $K$ (the number of items).
> 2. Their estimation bounds depend on the spectral gap of the random walk $\mu(P)$, which can scale as $1/(m \max_{i,j} (w_i/w_j)^2)$ (see Corollary 1). This introduces undesirable dependencies not only on  $m$ but also on $\max_{i,j} (w_i/w_j)^2$, which our approach explicitly aims to avoid.
> 3. The suggestion to add a reference arm is unclear—how should such a reference arm be chosen in practice? In contrast, our procedure uses a reference arm only in the analysis, not in the algorithm itself.
> 4. Finally, their method would still suffer from the limitation we address: requiring repeated display of the same set in long instances, which we aim to avoid.
>
> Please let us know if you have any additional questions. We will be happy to clarify.

---

> ### Comment · Reviewer_X5Yt · 2024-11-27
>
> I thank the authors for their response. I disagree with the authors conclusions on a few points. Regarding their comments about repetition vs non-repetition, I still maintain that an argument can be made one way or another, and both would remain valid unless one provides actual evidence supporting one over the other. Moreover, I still don't see repetition being a "weakness" since again, this online learning process isn't being done with a single user, rather a whole population, in which case repetition vs non-repetition becomes a rather meaningless discussion. I agree with the authors point about generalization of the no-choice assumption.
>
> Regarding their comments about using the MNL parameter estimation guarantees of Agarwal et al., ICML 2018, there are few potential misunderstandings on behalf of the authors. A lot of the pessimistic constants in their analysis appear due to the fact that they operate in the *passive data* regime, where one does not have control over the item sets that are compared. This problem on the other hand is in the *active learning* setting, where you can choose what item sets are compared, and as a consequence, have direct control over the topology of the underlying comparison graph, and you can choose sets such that the underlying comparison graph has a constant spectral gap, which is why I said using their method might give you much better dependence on problem constants. If you actually look at the proof of Lemma 7, where the spectral gap is bounded, the dependence is actually much weaker before it gets crudely bounded by the squared terms you mention in your response since there is no control over what the underlying graph may look like, hence they have to assume the worst possible case. Your rank breaking algorithm also has a rather suboptimal dependence on this ratio ($\theta_{\max}^{3/2}$ and $\theta_0$ is assumed to be 1 so its essentially the same as having a $w_{\max}/w_{\min}$) type dependence. Their algorithm actually has a dependence on something like $\max (w_id_i)/\min(w_id_i)$, which can give a potentially improved dependence on this ratio since $d_i$ is essentially the number of times item $i$ is compared, so in theory, if you scale $d_i \approx 1/w_i$, i.e. compare low weight items more than high weight items (which you can somewhat identify since low weight items will rarely win, and of course while trying to keep regret low) which again is possible to do since you get to choose what sets are played and how often, this dependence may go away completely. Moreover, since you can choose which items to compare, you can avoid having large weight items in the same comparison set as low weight items using say empirical selection probabilities to create these comparison sets (ideally, each comparison set should have items with comparable weights, and if you have really low weight items, then they should be identifiable and hence discardable quickly since they will be rarely selected). This would again circumvent this $w_{\max}/w_{\min}$ dependence, and would rather depend on this ratio only constrained to the items in the same comparison set, which would be a substantially weaker dependence than the global max to min weight ratio. In all, I am not convinced that this result is not already implied by a straightforward modification of existing work, and I am inclined to maintaining my original score.

---

> ### Author Response · Authors · 2024-11-27
> **Additional clarification for Reviewer X5Yt**
>
> Thanks for your comments. We would like to point out a few things that you might have missed, which might perhaps cause confusion.
>
> -(A) **Please note our main result is Thm. 5 and not Thm. 3:** Firstly, there is a great misunderstanding on the reviewer's part to identify our main result. *Please note, as we also clarified in our list of contributions (Sec 1) and in the Expts (Sec 5), our main result is Thm. 5 and not Thm. 3. Thus, as opposed to what you claimed, our result (Thm 5) is completely independent of the ratio $\theta_{\max}/\theta_0$* -- in fact, this is the key contribution of our work over MNL-Bandits (Agrawal et al'2019) which already establishes a regret bound of $\sqrt{\theta_{\max}/\theta_0 KT}$ regret bound, which is obviously vacuous in the regime of $\theta_{\max } \to infty$ keeping $\theta_0 = 1$ (or equivalently the regime where $\theta_{\max} >> \theta_0$). We clearly mentioned this advantage in several places, including Rem 1, Expts 1 (Averaged Regret with weak NC $\theta_{\max}/\theta_0 >> 1$), as well as in our list of contributions. Thus, please note our goal was to actually derive a $\theta_{\max}$ independent result, and we successfully did so with our Adaptive Pivot Selection algorithm in Sec 4, which provably runs in $\tilde O(K\sqrt T)$ regret (Thm 5). So your claim that
>
> > ``your rank breaking algorithm also has a rather suboptimal dependence on this ratio $\theta_{\max}/\theta_0$ ... so its essentially the same as having a $w_{\max}/w_{\max}$ type dependence" is incorrect and stands vacuous.
>
> -(B) **Our Algorithmic novelty lies far ahead of just the Rank Breaking (RB) idea:** *Secondly, note that our novelty not only lies in rank-breaking based estimation. There are several novelties behind our algorithmic idea that makes it regret optimal as well as computationally efficient*.  The novelty not only lies in our algorithmic ideas but also regret analysis which is equally interesting and involves non-trivial mathematical rigor. We explain this fully in the thread **Revisiting the key Ideas of our main algorithm (Adaptive AOA-RBPL, Sec 4) and its regret Analysis** above.  Further, owing to this novelty, *it is also easily extendable to general preference models, including RUM models as explained in Rem 2*, unlike the prior works or even the Spectral ranking method you described above.
>
> -(C)  **Hand wavy claims vs our Concrete proofs with solid empirical evidence**: Finally, with all respect, we firmly disagree with the reviewer that our result (Thm 5) can be obtained straighforwardly from the Spectral Ranking (Agarwal et al'2018) paper. *If it were so easy, we should have been able to derive it from a straightforward extension of their result, which is clearly not the case as coming out from this discussion conversation.*
>
> Here are the reasons why it is not straightforward. Firstly note our algorithm (Sec 4, Thm 5) is already independent of any such ratio term such as $w_{\max}/w_{min}$ as opposed to what you proposed for the derivation. Even if we ignore that for a moment and wish to derive a bound scaling with $\max(w_i,d_i)/\min(w_i,d_i)$, it is absolutely not clear how *``such dependency can potentially give an improved dependence on the ratio in the regret by setting $d_i \propto 1/w_i$''. **Firstly, (i)** $w_i$s are initially unknown to the learner, so a cost of estimation has to appear, which will contribute some multiplicative factor in the final regret guarantee. **Secondly, (ii)** if $w_{\max}/w_{\min}$ is small, this would yield a very high choice for the $d_i$ as it is set $\propto 1/w_i$ which would first increase the regret because of the dependence on the $d_i$. **Finally, (iii)** the computational complexity will shoot up, which will now need to construct many comparison sets, which will be a huge disadvantage compared to the minuscule computational overhead of our proposed algorithm in Sec 4. **And (iv)** *all these disadvantages exist even if we ignore the fact that our results are already independent of any ratio terms like $\theta_{\max}/\theta_0$ which your proposed algorithm will not be*.
>
> In summary, as theoreticians, we understand certain concepts might appear simple with informal or hand-wavy arguments without concrete derivations and proofs. However, the actual analysis often proves to be far complex and nuanced when formal algorithms and rigorous proofs are developed. If this was indeed so simple, it would likely have already been addressed and published, given the strong and practical motivation of our work (to generalize the results of MNL-Bandits by relaxing the strength of the NC item)—as admitted by the reviewer themself. We therefore respectfully urge the reviewer to kindly not undermine and disregard the novelty of our algorithm, analysis, and experimental studies in favor of a (hand-wavy) idea that has significant limitations, as we have outlined above.
>
> Pleasse let us know if you have any additional questions, would be happy to clarify.

---

> > ### Comment · Reviewer_X5Yt · 2024-11-27
> >
> > I thank the authors for their response. Regarding your comment about the regret term in theorem 5 being independent of the ratio $\theta_{\max}/\theta_{\min}$ is correct, but its not fully honest as you make it out to be in your rebuttal as it still has a dependence on $\theta_{\max}$, which is a weaker dependence, but present nevertheless. I agree with your assessment that modifying existing ideas in bandits + MNL parameter estimation to achieve this result is probably not as straightforward as I implied in my initial review, but I still strongly believe it is possible to achieve a similar result through the ideas I was outlining above. A lot of the pessimistic constants in the existing results arise purely due to the fact that these results operate in the passive data regime where we have no control over what item sets get compared, which will improve drastically in this bandit setting where we can choose what sets get displayed in every trial. Moreover, the objective in MNL parameter estimation is to estimate the entire parameter vector, whereas here, the objective here is much weaker, which is to accurately identify only the top-m items/ top-m revenue weighted items. Therefore, the estimation error for low weight items largely becomes irrelevant as they can be discarded early on in the process (if an item has very low weight, it will rarely win and hence, will quickly become distinguishable from the high weight items which can then be discarded). These together are what lead to my opinion that existing ideas in MNL estimation should give you this result with not too much effort. Due to the short review cycles at conferences, we get limited time as reviewers to make such assessments, and therefore, it cannot be expected for us to come up with entire formal proofs for our arguments. The best we can do is use our experience in related areas to judge the novelty of the paper being reviewed, what you call as handwaving. Too often do I see authors far too eager to reinvent the wheel instead of doing their due diligence with reviewing existing literature and assessing whether it is possible to achieve a result using existing results, which is not a good research practice. Given this discussion and general lack of mention of MNL parameter estimation literature in the paper, I am not certain that this isn't the case here. That being said, I am increasing my score to lean towards acceptance, though I would really like the involvement of other reviewers in this assessment.

---

> > > ### Author Response · Authors · 2024-11-28
> > > **Thank you**
> > >
> > > Thanks for your consideration; we really appreciate it.

---

### Official Review · Reviewer_KjYH · 2024-11-04

**Soundness:** 3
**Presentation:** 3
**Contribution:** 3
**Rating:** 6
**Confidence:** 2

**Summary:**

The paper revisits MNL bandit problems, and designs efficient algorithms for the problem of regret minimization in assortment selection. The authors propose a novel concentration guarantee for estimating the score parameters of the model using ‘Pairwise Rank-Breaking’, which builds the foundation of the proposed algorithm. The provided regret upper bound is quite comparable with state-of-the-art, with numerical simulations verifying their theoretical results.

**Strengths:**

* The paper proposed a new perspective to tackle the MNL bandit problem, which is interesting in the literature while sharing a quite comparable regret performance as Agrawal et al. 2019.

**Weaknesses:**

See Questions section.

**Questions:**

* In Line 315, I'm not sure if a regret bound of $\tilde{O}(\sqrt{KT})$ is optimal in the proposed model setting, as the lower bound of stochastic bandit is known as $\Omega(\log T)$ from Lai & Robbins (1985) [1], while $\Omega(T^{1/2})$ is known as the regret lower bound of adversarial bandit setting.
* How does the regret result relate to the maximum of assortment size $m$? In extreme case where $m=1$, the problem reduces to a standard bandit problem with a "no choice" action. How to understand the impact of $m$ on the regret?

[1] https://core.ac.uk/download/pdf/82425825.pdf

---

> ### Author Response · Authors · 2024-11-27
> **Rebuttal for Reviewer KjYH**
>
> Thanks for your careful reading and insightful comments. We address your questions below:
>
> **W1.** In Line 315, Optimality of the regret bound.
>
> Thanks for the question. You are right that stochastic multiarmed bandits have $\Omega(\log T)$ regret, but please allow us to first explain the difference between an instance (gap) dependent and instance independent (worst case) regret guarantee. This would clarify the confusion with adversarial and stochastic regret bounds. The standard K-armed multiarmed bandits (MAB) have two types of regrets
>
> - (i) $\Omega(\sum_{i =1 i \neq i^*}^K (\log T/\Delta_i + \Delta_i T))$ type instance (gap) dependent regret bound where $\Delta_i$ is the reward gap of the $i-th$ arm and $i^*$ is the the arm with the highest reward (the a.k.a. best arm).
>
> - (ii) $\Omega(\sqrt{KT})$ worst case (instance independent) regret guarantees. The name is justified since this regret bound is independent of the problem-dependent parameters ($\Delta_i$s). Note one can derive this regret bound from the above (instance dependent) lower bound for the (worst case) choice of $\Delta_i = \sqrt{K/T}, \forall i \in [K] \setminus \{i^*\}$ which indeed maximizes the instance-dependent lower bound in (i).
>
> The book Bandit Algorithms by Lattimore and Szepesvari (https://tor-lattimore.com/downloads/book/book.pdf) explains this very nicely in Chapter 13, 15 and 16. Please note both the above lower bounds apply to stochastic MAB problems, i.e. the underlying problem instance used towards proving the above lower bounds were stochastic K-armed MAB instances. In fact, even if you notice the lower bound derivation (see Thm 3) in Lai & Robbins (1985), as you pointed, the classical result essentially used a stochastic problem instance to derive the $\Omega(\sqrt{KT})$ lower bound.
>
> It is also worth noting that this means in terms of the regret objective, the problem complexity of adversarial MAB problems are no-harder than their stochastic counterpart (for the worst-case problem instances). Intuitively, the reason being this phenomenon is because  the classical K-armed MAB instance evaluates the regret of a learning algorithm against a fixed benchmark, even if the underlying problem instance is adversarial. Now since adversarial K-armed MAB is only a harder problem than stochastic K-armed MAB, the lower bound derived in (ii) is also a valid lower bound for adversarial K-armed MAB (Note the lower bound of (i) is ill-defined for adversarial MAB instances as there is no notion of gap $\Delta_i$s in those settings).
>
> Now, it is important to note that our problem, AOA-PL, is more general than K-armed MAB, and thus above lower bounds for the K-armed MAB might not directly apply for the problem for a general subsetsize m. However, the regret optimality of our method follows from the lower bound derived in Chen and Wang' 2017 as we mentioned in Line 353. The lower bound derived in Chen and Wang 2017 is $\Omega(\sqrt{KT})$, and it is of the nature of instance-independent worst-case lower bound, as explained in (ii) above. But it is a valid lower bound for our problem instance, in fact a tight one. It is also worth noting that deriving an instance (gap) dependent lower bound is much harder (compared to stochastic K-MAB) for our AOA-PL problem since the notion of gaps are different for our problem due to the different feedback model and regret objective. It is an open area of research which will require new mathematical rigor. Nevertheless, we show the regret performance of our proposed methods is tight in terms of the worst-case instance-independent lower bound as known by Chen and Wang's 2017. I hope that clarifies your questions.
>
> -- The next question is continued in the thread below --

---

> ### Author Response · Authors · 2024-11-27
> **Rebuttal for Reviewer KjYH (contd)**
>
> **Q2.** How does the regret result relate to the maximum of assortment size m? ... How to understand the impact of $m$ on the regret?
>
> Thanks for the great question! The answer is the problem complexity does not depend on the subsetsize $m$! --- Although it could be counter-intuitive but this is a known phenomenon with this class of MNL bandit problems. To see this mathematically, note the regret bound derived in Chen and Wang' 2017 is already independent of $m$ as we explained above in **Q1**. Further proof of tightness comes from the $\tilde O(\sqrt{KT})$ matching upper bound results which establishes the tightness of the regret lower bound. Intuitively, this could be explained by the logic explained in the Remark 2 Saha & Gopalan'2019 (https://proceedings.mlr.press/v98/saha19a/saha19a.pdf), which explains that
>
> - *Surprisingly ... the problem complexity of identifying a near-optimal item from m-sized subsets does not reduce with m, implying that there is no reduction in hardness of learning from the pairwise comparisons case (m = 2): On one hand, one may expect to see improved performance as the number of items being simultaneously tested in each round is large ($m$). On the other hand, the performance could also worsen, since it is intuitively ‘harder’ for a good (near-optimal) item to win and show itself, in just a single winner draw of $m$ items against a large population of $m$ − 1 other competitors which increase the variance in the regret model and making the learning problem harder. The result, in a sense, formally establishes that the former advantage is nullified by the latter drawback making the final regret performances independent of the subsetsize $m$*.
>
> Also, when m = 1, the learner is supposed to play subsets $S_t$ of size $1$, which means they only get to select one item that competes with the NC item every round (note NC is never counted in the subsetsize $m$ as it is the default item present in every set). In this case again, the regret learning tradeoff remains the same and independent of $m$ following the same logic and intuition we provided above.
>
> More interestingly, as a side note it is worth pointing out that, if you note our experiments in *(3) Averaged Regret vs Length of the rank-ordered feedback ($k$), Sec 5*, we show that, if we make the feedback model of Eqn (1) richer with $k$-length rank-ordered feedback, where at every round upon playing the $m$-sized subset $S_t$, the learner gets to see the top-$k$ draws ($k \leq m$) from the PL model without replacement, then the regret performance indeed improves with increasing $k$, as also intuitive. *Our Figure 3 in Sec 5 justifies this tradeoff*. This also corroborates the conclusion of *Remark 6 of Saha & Gopalan'2019*, which claims that how performance improves as the learner has access to larger length rank-ordered results compared to just the winner feedback in the PL model considered in Eq (1), which is equivalent to k=1. I hope that clarifies your question.
>
> Thanks again for the questions; we will definitely add comments to clarify both justifications for Q1 and Q2.
>
> Please let us know if you have any additional questions. We will be happy to clarify.

---

### Official Review · Reviewer_t5dy · 2024-11-13

**Soundness:** 3
**Presentation:** 2
**Contribution:** 3
**Rating:** 6
**Confidence:** 3

**Summary:**

This paper revisits the Active Optimal Assortment (AOA) problem, and proposes two algorithms: $\mathrm{AOA-RB_{PL}}$ (Algorithm 1) and $\mathrm{AOA-RB_{PL}-Adaptive}$ (Section 4). Regret bounds have been established (Theorem 3, 4, 5) and preliminary experiment results have been demonstrated in Section 5.

**Strengths:**

- To the best of my knowledge, the proposed algorithms in this paper are novel, and they are based on some new insights like Rank-Breaking (proposed in Khetan & Oh (2016)) and Adaptive Pivot Selection (proposed in this paper).

- The analyses in this paper seem to be rigorous and highly non-trivial. I have not checked the analyses line by line, but the derived regret bounds make sense to me.

- The authors have done a good job of literature review and have well positioned this paper with respect to relevant literature.

**Weaknesses:**

- The writing of this paper can be further improved. Section 3 and Section 4 are too technical and a little bit hard to read.

- All the experiments in this paper are synthetic. Might the authors add some experiment results based on real-world datasets?

- [typo] for Arith50 environment, $\theta_i = 1-(i-1) \times 0.2$ for $i \in [50]$. This seems to be a typo since $\theta_i$ can be negative.

**Questions:**

Please address the weaknesses above.

---

> ### Author Response · Authors · 2024-11-26
> **Rebuttal for Reviewer t5dy**
>
> Thanks for your careful reading and insightful comments. We address your questions below:
>
> **W1.** The writing of this paper can be further improved. Section 3 and Section 4 are too technical and a little bit hard to read.
>
> -- Thanks so much for the suggestion; we agree Sec 3 and 4 could have been less technical, we will either omit the proof sketches fully or shorten them significantly and add more intuitive/textual description of the math equations in the extra space. It will also improve the readability and flow of the text. Thanks.
>
> **W2.** All the experiments in this paper are synthetic.
>
> -- Thanks for your comment. Please note we are unaware of any open-sourced real dataset for the problem that can be used to validate our algorithms for these particular problem settings. One of the challenges being since it is impossible to know the ground truth parameters $(\boldsymbol{\theta})$ of the MNL model, it is hard to evaluate the regret performance of the algorithms. This is also the reason why most of the existing works only reported synthetic data experiments (Agrawal et al'17, Leslie & Grant'23). Although Agrawal et al'19 claim to have used the `UCI Car Evaluation Dataset', however, in reality, they synthetically modified the item features and assigned synthetic scores to the PL parameters through a linear score transformation (pls see Sec 7.3 of Agrawal et al'19) which ultimately only results into a synthetic data experiment only. Thus we are unable to include any real-world experiments due to the lack of any suitable open-sourced dataset available for the MNL-AOA setting.
>
> Further, since the main focus of our work has been primarily theoretical, we conduct synthetic experiments to validate our main theoretical guarantees. However, if you are aware of any suitable dataset, we will be happy to report the comparative performance of the baselines on that dataset. Please let us know if you have any suggestions. Thanks for your feedback.
>
> **W3.** [typo] for Arith50 environment.
>
> -- Thanks so much for pointing this out. Indeed, it should have been $\theta_i = 1 - (i-1)\times 0.02, ~\forall i \in [50]$, which results in $\theta_i$s to be positive. *We have already corrected this in the updated draft*, please advise if you suggest any further changes. Thanks again.
>
>
> Please let us know if you have any additional questions. We will be happy to clarify.

---

### Author Response · Authors · 2024-11-27
**[In response to the questions raised by Reviewer X5Yt] Revisiting the key Ideas of our main algorithm (Adaptive AOA-RBPL, Sec 4) and its regret Analysis**

We gave two algorithms, (**Alg1**) AOA-RBPL in Sec3, and (**Alg2**) AOA-RBPL-Adaptive, in Sec4, both of which use two different estimates of $\{\theta_{i,t}^{ucb}\}_{i \in [K]}$, $\theta_{i,t}^{ucb}$ being the upper confidence bound (UCB) estimate of the MNL parameter $\theta_i$ at round $t$, for all $i \in [K]$. *Due to the scale independence of the MNL model, we always assume that the parameter of the no-choice (NC) item is always $\theta_0 = 1$* (see Line 204).

**How $\boldsymbol{\theta}^{ucb}$ was used in the regret analysis (Thm 3, Thm5):** We will explain the intuition of $\theta_{i,t}^{ucb}$ for Alg1 and Alg2, but before that let us understand how $\theta_{i,t}^{ucb}$ were used in the regret analysis of Thm3 and Thm5 (resp. for Alg1 and Alg2):

- (1) Towards this, an important observation to note is both our regret analyses (i.e. proof of Thm5 and Thm6) uses a **key wtd-utility inequality** $\mathcal R(S^*, \boldsymbol{\theta}) \leq  \mathcal R(S^*, \boldsymbol{\theta}^{ucb})$, where $\mathcal R(S^*, \boldsymbol{\theta})$ and $\mathcal R(S^*, \boldsymbol{\theta}^{ucb})$ respectively denote the weighted-utility of set $S^*$ under MNL parameters $\boldsymbol{\theta}$ and $\boldsymbol{\theta}^{ucb}$ (see Eq2): Please see the in proof of Thm3 and Thm6 respectively, to note how we used the above key inequality to derive the final regret upper bound ($Reg_T^{wtd}$) for Alg1 and Alg2).

- (2) However, to achieve the above property we need to ensure, $\theta_i^{ucb} \geq \theta_i$ as we showed in Lem8: Essentially Lem8 shows that if $\theta_i^{ucb}$ is a valid UCB of $\theta_i, ~\forall i \in [K]$, then the estimated wtd-utility $\mathcal R(S^*, \boldsymbol{\theta}^{ucb})$ is also a valid UCB of $\mathcal R(S^*, \boldsymbol{\theta})$.

- (3) Hence the question is how to assign the values of $\theta_i^{ucb}$ so that they represent a valid and tight UCB of the corresponding (true) MNL parameters $\theta_i$s?

We will now justify our choice of $\theta_i^{ucb}$s for all $i \in [K]$ for which the above properties are satisfied, both for Alg1 and Alg2. But let us understand an important property of the MNL model before that.

**$\boldsymbol{\theta}$ estimate from MNL pairwise preferences:**

- (1) Thanks to the property of the MNL model, we first note that $\textbullet $ For any two items $i, j \in [K]\cup \{0\}$, the pairwise preference of $i$ over $j$ is $p_{ij} = \frac{\theta_i}{\theta_i + \theta_j}$ by definition (see Eq1).

- (2) But since $\theta_0 = 1$ (Line204), this implies $\textbullet (ii) ~\theta_{i} = \frac{p_{i0}}{1 - p_{i0}}, ~\forall i \in [K]$.

- (3) *However $p_{i0}$ is unknown, but can we estimate it?* Here we have a **key idea** that by exploiting the *Independence of Irrelevant Alternatives (IIA)* property of MNL model once can indeed maintain unbiased pairwise preference estimates ($p_{ij}$s) of any pair $(i,j)$, $i,j \in [K]\cup\{0\}$ using Rank-Breaking. We denoted them by $\hat p_{ij,t}$ at round $t$ (Eq.3).

Let us now first understand the rationale behind our choice of $\boldsymbol{\theta}_t^{ucb}$ for Alg1.

> $\boldsymbol{\theta}_t^{ucb}$ justification for Alg1 (uses the NC item as pivot):

- (1) Upon realizing $\theta_i = \frac{p_{i0}}{1 - p_{i0}}$ and having access to $\hat p_{i0,t}$, the unbiased estimates of $p_{i0}$ derived through rank breaking as described above, we first find a **good upper confidence bound** (UCB) estimate of $p_{i0}$, denoted by $p_{i0,t}^{ucb}$ as described in Eq3. Further, we derive the concentration rate (tightness) of the UCB estimate $p_{i0,t}^{ucb}$ in Lem7 (Appendix B.1).

- (2) The next idea was to use $p_{i0,t}^{ucb}$ to define a natural estimate of $\theta_{i,t}^{ucb} = \frac{p_{i0,t}^{ucb}}{[1 - p_{i0,t}^{ucb}]+}$ (see the display after Eq3, we are unable to construct the right display with the openreview markdown), drawing inspiration from $\theta_i = \frac{p_{i0}}{1 - p_{i0}}$. Further, Lem1 establishes the rate of concentration of the UCB estimate $\theta_{i,t}^{ucb}$ using Lem8 (see proof of Lem1 and 7 in Appendix B.1, B.2).

This concludes our intuition behind our choice of $\theta_{i,t}^{ucb}$, which provably yields a UCB estimate of the true MNL parameters $\theta_i$ for all $i \in [K]$.

---

> ### Author Response · Authors · 2024-11-27
> **Contd: [In response to the questions raised by Reviewer X5Yt] Revisiting the key Ideas of our main algorithm (Adaptive AOA-RBPL, Sec 4) and its regret Analysis**
>
> > While Alg1 carves out the basic building block of our $\theta_{i,t}^{ucb}$, one caveat lies in its concentration rate which shrinks at the rate of $O(1/\sqrt{n_{i0,t}})$ --- as established is Lem7, $n_{i0,t}$ being the number of rank-broken pairs of $(i,0)$ till time $t$. Thus ideally, we would need $n_{i0,t}$ to grow fast for a fast and tight convergence of $\theta_{i,t}^{ucb}$ to $\theta_i$. However, as Lem2 reflects, this might not be true unless either $0$ (NC) or $i$ is a `strong item' with sufficiently large MNL parameters (comparable to $\theta_{\max}$). This is since, as Lem2 shows $n_{i0,t}$ (roughly) grows proportional to $\frac{(\theta_i + \theta_0)}{\theta_{\max}}$ which could be quite small if both Item-i and 0 (NC) happens to be a "weak-items" in terms of their MNL scores, i.e. $\max{\theta_i,\theta_0} << \theta_{\max}$.
>
> If we think, this is also intuitive since, if both $\theta_i$ and $\theta_0 = 1$ are small compared to $\theta_{\max}$, chances are very low that either of them will be picked as a winner of any round $t$, even if $i \in S_t$ and as a result they will never be "rank-broken" against each other resulting in a very small value of $n_{i0,t}$, weaker UCB estimates $\theta_{i,0}^{ucb}$ and finally a weighted regret bound of $Reg_T^{wtd} = \tilde O(\sqrt{\theta_{\max} KT})$ (Thm3) which could be large when $\theta_{\max}$ is large!
>
> Understanding the problem, we remedy this with Alg2 (AOA-RBPL-Adaptive) in Sec4, where we devised a smarter UCB estimate of $\boldsymbol{\theta}$, while still keeping the basic intuitions from Alg1 (AOA-RBPL) intact:
>
> > $\boldsymbol{\theta}t^{ucb}$ justification for Alg2: As explained above, we realized "pivoting" on the NC for estimating $\theta_i$ could be a bad idea, especially if $1 = \theta_0 << \theta{\max}$. Towards this we made the following (seemingly interesting) observations:
>
> - (1) We first note that: $\theta_i = \frac{\theta_i}{\theta_0} = \frac{\theta_i}{\theta_j}\frac{\theta_j}{\theta_0}$ for any $j \in [K] \setminus {i}$.
>
> - (2) Then drawing motivation from the UCB estimates of Alg1, we further set $\theta_{i,t}^{ucb} = \gamma_{ij,t}^{ucb}\gamma_{j0,t}^{ucb}$, where $\gamma_{ij,t}^{ucb} = \frac{p_{ij,t}^{ucb}}{(1 - p_{ij,t}^{ucb})_+}, ~\forall i,j \in [K]\cup{0}$ (display of Line428).
>
> - (3) The hope is if we can find a "strong element j" and pivot our rank-broken pairwise estimates ($p_{ij,t}^{ucb}$s) around $j$ for all $i \in [K] \cup {0}$, hopefully the that will remedy the caveat of Alg1 as detailed above.
>
> - (4) In order to find such a "strong pivot j" (such that $\theta_j$ is comparable to $\theta_{\max}$), we set a dynamic $j = \arg\min_{j \in [K] \cup{0}} \gamma_{ij,t}^{ucb} \gamma_{j0,t}^{ucb}$ for each item $i \in [K]$ and time $t$, which by definition happens to be a relatively stronger item than Item-$i$ and 0 (NC).
>
> - (5) Further, similar to Lem1, we also find the rate of concentration of our new UCB estimates $\theta_{i,t}^{ucb}$ in Lem9 (see Appendix B.6), which, as intuitive, is shown to shrink at the rate of $O(\frac{1}{\sqrt{n_{i,j,t}n_{j0,t}}})$. The nice trick was to note that this yields a sharp concentration for $\theta_{i,t}^{ucb}$ owing to our clever choice of the dynamic pivot j as described in the point above -- this saves us from the caveat arose in Alg1 due to a poor choice of the static NC pivot (Item-0).
>
> - (6) Finally the above tight and sharp concentration of $\theta_{i,t}^{ucb}$ in Lem9 ensures the final weighted regret of $Reg_T^{wtd} = \tilde O(\sqrt{\min{\theta_{\max},K} KT})$ (Thm5) --- Note this could be at most $\tilde O(K\sqrt T)$ even if $\theta_{\max} \to \infty$. Here lies the drastic improvement of our algorithm compared to prior works listed in Table 1 which either needs to assume $\theta_{\max} = \theta_0 = 1$ or their regret scales as $O(\sqrt{\theta_{\max}KT})$ leading to a trivial regret bound (as $\theta_{\max} \to \infty$ or even is $\theta_{\max} = \Omega(T)$). We discussed this in detail in Rem1 after Thm5, as well as empirically validated in Sec5.
>
> We hope that was helpful. Please let us know if you need further explanations.
>
> Thanks for your attention.
> Authors

---

### Meta-Review · Area_Chair_eaxd · 2024-12-23

**Metareview:**

The paper studies  Active Optimal Assortment (AOA) problem under MNL bandit feedback. The authors proposes two algorithms based on a  concentration guarantee for estimating the score parameters of the model using ‘Pairwise Rank-Breaking’. Theoretical analysis of regret bounds and synthetic experiments validated the effectiveness of proposed methods. The reviewers recognized the following strengths:
- The proposed algorithms are novel and intuitive.
- Theoretical analyses are sound.
- Paper organization and presentation are clear.

The reviewers raised several questions and concerns:
- Technical novelty and challenge, especially comparing to deriving similar results by combining existing techniques.
- Experiments are synthetic, lack of experiments on real-world data.
- Clarification questions regarding motivating examples, regret bounds, comparison to other approaches.

Three reviewers are positive about the paper after discussion with the authors and one reviewer did not engage in discussion. The key remaining concern is whether we can derive similar results by combining existing techniques raised by Reviewer X5Yt. I see this as a valid question and thank the authors and the reviewer for multi-round thorough discussion. I recommend acceptance considering the paper's strengths and reviewers' positive opinions, and strongly suggest the authors to include detailed discussion regarding this issue in final version.

**Additional Comments On Reviewer Discussion:**

The reviewers raised several questions and concerns:
- Technical novelty and challenge, especially comparing to deriving similar results by combining existing techniques. After multi-round engagement, the response did not fully resolve Reviewer X5Yt's concern. Additional discussion should be included in final version to address the question.
- Experiments are synthetic, lack of experiments on real-world data. The authors explained the challenge of using real-world data in this setting, answering the question raised by Reviewer t5dy and Jq9h.
- Clarification questions regarding motivating examples, regret bounds, comparison to other approaches. IMO these questions are address in rebuttal.

---

### Decision · Program_Chairs · 2025-01-22

Accept (Poster)